# Optogenetic manipulation of cell migration with high spatiotemporal resolution using lattice lightsheet microscopy

Wei-Chun Tang [1], Yen-Ting Liu [1], Cheng-Han Yeh[1], Chieh-Han Lu [1], Chiao-Hui Tu[1], Yi-Ling Lin[2,3], Yu-Chun Lin [4,5], Tsui-Ling Hsu[6], Liang Gao[7], Shu-Wei Chang [1], Peilin Chen [1,8✉] & Bi-Chang Chen [1✉]

Lattice lightsheet microscopy (LLSM) featuring three-dimensional recording is improved to manipulate cellular behavior with subcellular resolution through optogenetic activation (optoLLSM). A position-controllable Bessel beam as a stimulation source is integrated into the LLSM to achieve spatiotemporal photoactivation by changing the spatial light modulator (SLM) patterns. Unlike the point-scanning in a confocal microscope, the lattice beams are capable of wide-field optical sectioning for optogenetic activation along the Bessel beam path.We show that the energy power required for optogenetic activations is lower than 1 nW (or 24 mWcm$^{-2}$) for time-lapses of CRY2olig clustering proteins, and membrane ruffling can be induced at different locations within a cell with subcellular resolution through light-triggered recruitment of phosphoinositide 3-kinase. Moreover, with the epidermal growth factor receptor (EGFR) fused with CRY2olig, we are able to demonstrate guided cell migration using optogenetic stimulation for up to 6 h, where 463 imaging volumes are collected, without noticeable cellular damages.

[1] Research Center for Applied Sciences, Academia Sinica, Taipei 11529, Taiwan. [2] Institute of Biomedical Sciences, Academia Sinica, Taipei 11529, Taiwan. [3] Biomedical Translation Research Center, Academia Sinica, Taipei 11529, Taiwan. [4] Institute of Molecular Medicine, National Tsing Hua University, Hsinchu 30013, Taiwan. [5] Department of Medical Science, National Tsing Hua University, Hsinchu 30013, Taiwan. [6] Genomics Research Center, Academia Sinica, Taipei 11529, Taiwan. [7] Key Laboratory of Structural Biology of Zhejiang Province, School of Life Sciences, Westlake University, Hangzhou, Zhejiang 310024, China. [8] Institute of Physics, Academia Sinica, Taipei 11529, Taiwan. ✉email: peilin@gate.sinica.edu.tw; chenb10@gate.sinica.edu.tw

Spatiotemporal manipulation and recording of biological processes, such as cell migration or membrane dynamics with the subcellular resolution, is crucial in understanding these fundamental cellular processes[1–3]. However, it is technically challenging for cell manipulation with a precise spatiotemporal resolution, especially in three dimensions (3D)[4, 5]. A common approach for subcellular manipulation can be achieved by optogenetic tools on a conventional microscope such as wide-field or confocal microscope[6–8]. Recording the 3D rapid cellular response at high spatiotemporal resolution can be problematic, where a high numerical aperture (NA) objective is often used to create a tightly focused spot for better confinement of the photoactivation beam. When such a photoactivation beam propagates in the cell, it activates molecules not only in the focal spot but also along the beam path where the out-of-focus activation may not be imaged when a pinhole is used to reject the out of focus signal as in the case of confocal microscopy (the beam paths of different excitation and detection schemes can be found in Fig. S1)[9]. As a result of tight focusing, an uneven photo-excitation along the beam propagation is created. Therefore, it is desirable to develop a wide-field technique with good sectioning capability to record the optogenetic response. A notable implementation in this regard is the development of lattice lightsheet microscopy (LLSM), which is constructed by an array of Bessel beams, where destructive interference of the coherent Bessel beams minimizes the contribution from sidelobes within the detection zone[10]. LLSM provides several advantages for high-resolution volumetric imaging of living cells, including reduced phototoxicity in which the excitation energy is distributed over an array of the illumination beams, an optimal balance between the thickness and length of the lightsheet, enabling high spatial resolution imaging with a good optical sectioning capability, and improved signal-to-noise ratio in which the out-of-focus noise is suppressed by the coherent modulation of lattice beams[11–13]. In addition, the lattice patterns also allow easy implementation of super-resolution microscopy using structured illumination[14].

With the help of LLSM, several biological processes, including the dynamics of the immunological synapses, transcription factors, and lymphocyte migration, have been investigated in great detail[15–17]. With an additional optical path for illumination, it has been demonstrated that photoablation/FRAP/photoactivation experiments[18–20] can be conducted on LLSM. However, it remains challenging to perform experiments requiring optical stimulation, such as subcellular optogenetic applications, in LLSM. In this situation, the lattice beams illuminate the samples at a tilted angle, which results in poor spatial activation of the signaling molecules within a cell, especially along the lightsheet propagation direction. LLSM could be potentially useful in living cells to conduct long-term optogenetic experiments with a very high spatiotemporal resolution at very low phototoxicity if the light stimulation can be integrated into LLSM. In this study, we report the use of LLSM for optogenetic experiments, called optoLLSM. The optical configuration of optoLLSM is the same as the LLSM reported previously. For optoLLSM, the lattice beams is used to monitor cellular behavior, whereas optogenetic molecules are activated by a Bessel beam, whose 3D energy distribution can be precisely controlled by the phase profiles on the SLM. Optically-induced clustering and membrane ruffling, in addition to guided cell migration, can be achieved in optoLLSM.

To demonstrate the performance of optoLLSM, we conducted several cell-migration-related optogenetic experiments. The traditional approach to inducing cell migration is to create concentration gradients of the growth factors[21] through diffusion mimicking the native cell migration environment. However, the diffusion gradient is difficult to maintain for precise cell manipulation at the single-cell level. The newly developed optogenetic

tools enable cell manipulation through light, where cells are modified genetically to respond to a specific wavelength of light[22, 23]. The most common genetic modifications for optogenetic experiments include optically induced heterodimerization (cryptochrome 2-calcium and integrin-binding N-terminal domain [CRY2-CIBN])[24] and optically induced clustering (CRY2olig)[25]. In this experiment, we adapted both systems to control cellular behavior. To achieve 3D cellular manipulation in optoLLSM, we demonstrated that our approach offers the advantages over original LLSM including controllable subcellular optogenetic activation and long-term cell manipulation.

## Results

**Bessel beam activation in lattice lightsheet microscopy for optogenetics.** The optoLLSM consists of a large parallel array of coherently interfering Bessel beams through the SLM in which individual Bessel beams with selected wavelengths can be used as activation sources by simply changing the patterns on the SLM[26] located at the plane conjugated to the imaging plane (Fig. 1a, b). With a different pattern on the SLM, the lattice beams at the imaging plane can be switched to a stimulating Bessel beam whose x and y positions can be controlled by different modulations in the SLM patterns, as shown in Fig. 1b. In addition, the lattice beams can be dithered to create a uniform excitation sheet to obtain 3D images of cells with the sample scanning mode, as shown in Fig. 1c. To demonstrate the advantages of using Bessel beam excitation for optogenetic applications, we developed two probes based on the CRY2 system: (1) CRY2olig and (2) CRY2-CIBN. In the optically-induced clustering probe, we fused CRY2olig with a mRuby3. As the CRY2 molecules oligomerize upon blue light illumination, the mRuby3 fluorescence proteins move closer together, forming a bright emitting spot (Fig. 1d). In this experiment, we tested several activation powers ranging from 0.5 to 4 nW in which the Bessel beam was formed using maximum and minimum numerical apertures (NAs) of 0.64 and 0.56, respectively. The theoretical and experimental cross-sectional intensities of the Bessel beam along the beam propagation direction are illustrated in Fig. S2 of the Supplementary Information. The time-lapse maximum intensity projection (MIP) images of the CRY2olig-mRuby3 transfected human bone osteosarcoma epithelial cell line (U2OS) with various stimulation energies are shown in Fig. 1e. At each time point, a series of images were collected in which the cell was first excited by a shorter wavelength laser for activation and imaged by LLSM with a 10 ms exposure time for both activation and imaging. The cells were then moved to the next position, and the process was repeated until the volumetric image was collected. Each imaging volume consisted of 131 planes, which took 2.6 s to collect. After optical activation, a significant aggregation of CRY2olig-mRuby3 in the middle area of the cells illuminated by the Bessel beam could be found (Fig. 1e and Supplementary Movie 1). Since the sample scan mode was used, lines of mRuby3 fluorescence signals could clearly be seen at the center of the cells. The diffusion of CRY2olig-mRuby3 into the cytoplasm caused a broadening of the width of the observed CRY2olig-mRuby3 fluorescent strips over time. The intensity profiles of different Bessel beams formed by various maximum and minimum NAs in addition to the corresponding time-lapse MIP images of cell responses at the excitation laser power of 1 nW (~24 mWcm$^{-2}$) are shown in Fig. S3 of the Supplementary Information. Note that the minimum excitation power at the focal plane required to produce a visible clustering signal was 0.2 nW, which was low enough for a long-term living cell experiment. Before stimulation, no clustering was observed, indicating that the 561 nm laser used for volumetric imaging in the LLSM could not trigger the clustering of the

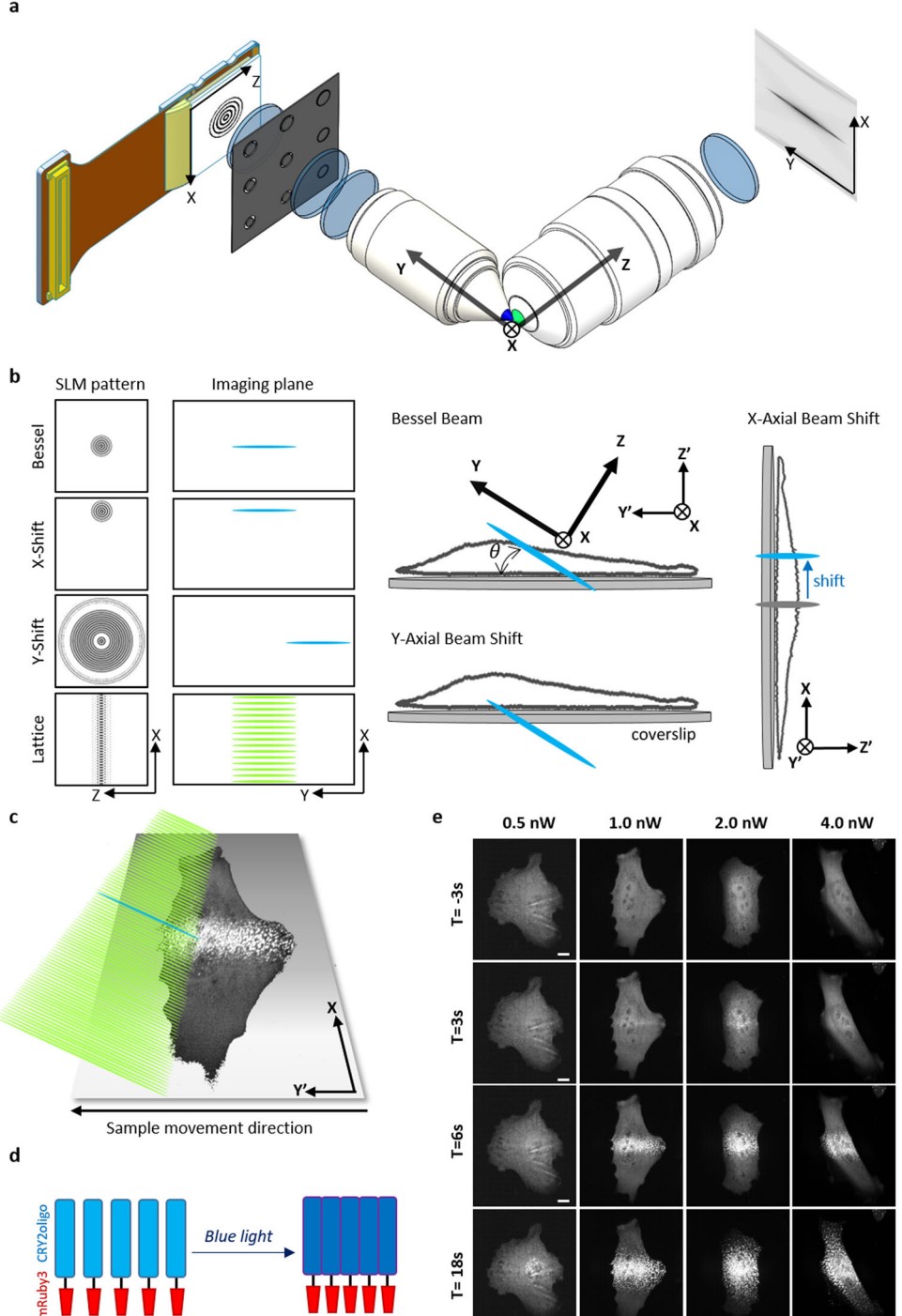

**Fig. 1 The scheme for optoLLSM and light-triggered CRY2olig aggregations. a** The schematic diagram of the major optical components in lattice lightsheet microscopy (LLSM). The patterns displayed by the spatial light modulator (SLM) are imaged at the pupil plane of the excitation objective through an annual mask and lens. Lattice beams, in addition to Bessel beams with different X, and Y positions, can be generated at the imaging plane by imposing different patterns on the lightsheet microscopy (LSM). **b** Various patterns on the SLM for generating the corresponding excitation beams in the imaging plane. The position of the Bessel beams can be shifted in the X or Y direction by using X- or Y-shift patterns. For normal Bessel beam stimulation, the Bessel beam with a length of 20 μm illuminates the cell at a tiled angle of 32.8 degrees. With Y-shift stimulation, the Bessel beam can be moved along the propagation direction to illuminate the basal membrane. With an X-shift pattern, the Bessel beam can be moved to a different location in the x-direction with an accuracy that is better than 100 nm. **c** The illumination scheme for an optogenetic experiment. The blue beam represents the Bessel beam at 488 nm used to activate the optogenetic molecules, and the green lightsheet represents the 561 nm lattice lightsheet for imaging. The volumetric image was obtained by dithering the lattice beam and moving the sample along the Y' direction. **d** The schematic for optical induced clustering of cryptochrome 2 (CRY2)olig-mRuby3. Blue: CRY2olig, Red: mRuby3. **e** The time-lapse maximum intensity projection (MIP) images of CRY2olig-mRuby3 expressed cells with various activation energies. The interval between each stack is 3 s. Scale bar 10 μm.

CRY2olig-mRuby3 protein (row 1 of Fig. 1e). We also tested the activation using different wavelengths (405, 445, 488, and 514 nm) at 1 nW. The activation of CRY2olig-mRuby3 was observed for all of the tested wavelengths (Fig. S4 and Supplementary Movie 2 in the Supplementary Information).

**3D activation using shifting Bessel beams**. A Bessel beam is a non-diffracted beam in which the energy concentrates within its length and diverges rapidly, as shown in Fig. S1. Taking advantage of this property, we demonstrated that it was possible to shift the lattice beam for fast and large volumetric imaging, during which the lattice beam could be repositioned in the propagation direction by imposing a defocusing phase profile on the SLM[27]. Therefore, it may be possible to create a Bessel beam that can locally activate molecules in the beam propagation direction. In the previous section, we demonstrated that the Bessel beam could activate the CRY2olig-mRuby3 at a specific location in each 2D frame during volumetric imaging, where the contineous activation at the same postion in diffrernt frames by Bessel beams acted as a Bessel fan activation on the Y'Z' plane within one volume scan (Fig. 2a, Bessel fan). The corresponding timing diagrams for switching on the activation (blue) and imaging (green) laser by two-color SLM and the movement of the sample stage (black) are illustrated at the bottom of Fig. 2a. To control the activation in the beam propagation direction, we generated a shifted Bessel fan in the Y direction via SLM with the phase map shown in Fig. 1b. The shift distance could be adjusted by changing the amplitude of the modulation phase profile on the SLM to locally activate the basal area of the cell (Fig. 2a, shifted Bessel fan) with the defocusing Bessel pattern for activation using the same timing diagram. Projection images of the cells activated by the Bessel fan activation across or underneath the cell body at different time points are shown in Fig. 2b (Supplementary Movies 3 and 4). The depth color image was used to illustrate the clustering of molecules and the cell edge. For the regular Bessel fan activation, the XZ' and Y'Z' projections (first row in Fig. 2b) indicate that the clustering CRY2olig-mRuby3 initially took place 3 s after stimulation in the middle of the cell. In contrast, the clustering of CRY2olig-mRuby3 was observed 21 s after activation at the bottom of the cell when a shifted Bessel fan was used (second row in Fig. 2b).

To further demonstrate the capability of optoLLSM to conduct 3D activation, we first obtained a volume image of the cell without activation to select the proper frame for activation. With the help of additional clock device in LLSM, the single Bessel beam activation could be achieved by switching activation wavelength 488 nm at a given frame within a 3D stack scan (Fig. 2a single Bessel Beam) where the timing diagram indicates the activation beam is switched on once at a specific z step during volumtric imaging. Unlike the Bessel fan stimulation in which the cells were stimulated in every frame, the single Bessel beam stimulation only activated the cells once per volume scan. The MIP images of the cells activated by such single Bessel beam stimulation are depicted in Fig. 2c, in which the clustering of CRY2olig-mRuby3 in the illuminating region could clearly be seen 21 s after stimulation. Our results indicate that the clustered fluorescent proteins slowly diffused away from the illuminated area (marked as a white cross in Fig. 2c and Supplementary Movie 5). To analyze the spatiotemporal behavior of the activated clusters by single Bessel beam activation, 3D cell volume renderings with selected CRY2olig-mRuby3 clusters are shown in Fig. 3a, where cluster 1 is located right at the activation region, clusters 2 and 3 are 2 μm and 5 μm away from the activation region, respectively. The time-dependent fluorescence intensity changes of the chosen clusters are plotted in Fig. 3b, where cluster

1 shows an instantaneous increase upon the activation beam. The delayed responses are observed according to the distance from the activation spot, for example, 10 s for cluster 2 and 40 s for cluster 3. To evaluate the photobleaching effect in our system, we monitored the fluorescence signal of a pre-existing cluster (cluster 4), which was far away from the activation beam. No decrease in fluorescence intensity of cluster 4 was observed during the whole experimental period indicating that the photobleaching effect of our system is negligible. A series of raw images for the selected plane activated by a single Bessel beam is shown in Fig. 3c, where the formation of the cluster over time is marked in a white box. The analysis of fluorescence intensities of different clusters activated by Bessel fan and shifted Bessel beam is shown in Figs. S5 and S6. These findings clearly demonstrate that optoLLSM could conduct an optogenetic experiment with good spatial control using a regular or modified Bessel fan/beam as an activation source.

**Subcellular optogenetic activation of membrane ruffling**. To control cellular responses at the subcellular level, we adapted the CRY2-CIBN system to trigger the regional cell membrane ruffling. In this experiment, the CRY2 was fused with PI3K-catalyzed motif (p85, iSH), which could trigger membrane ruffling by enhancing Rac1 and cdc42 signaling for actin filament elongation[28–30], whereas a plasma membrane-anchor peptide (CAAX motif) was attached to CIBN. To observe membrane ruffling, F-tractin-mCherry was used to monitor cytoskeletal dynamics. A 2A-self cleaved peptide (p2a) was used to connect each component, including F-actin-mCherry, CRY2-iSH, and CIBN-CAAX to ensure nearly equal expression levels in the cell. We first tested CRY2-mCherry-iSH-p2a-CIBN-EYFP-CAAX transfected cells using a Bessel fan activation at 488 nm to trigger the response in the selected areas[31]. The MIP images of CRY2-mCherry in the cell in XY' plane are shown in Fig. S7 and Supplementary Movie 6 in the Supplementary Information. To demonstrate that membrane ruffling can be induced with subcellular resolution, two areas within a cell that were separated by 15 μm were illuminated sequentially by the activation beam (regions 1 and 2 in Fig. S7), and membrane ruffling at each location could clearly be seen through quantification by fluorescent signals after stimulation. To evaluate whether the function of the cells was compromised by the activation process, we waited 30 min for the cell to recover to the non-activated state (as shown in Supplementary Movie 6). A lattice beam at 488 nm instead of a Bessel fan was used to activate the whole cell. Membrane ruffling was observed over the whole-cell, unlike the local membrane ruffling shown in Fig. S7, indicating the function of the cell remained the same even after activation by the Bessel fan in two subcellular locations. Since CRY2 moved from the cytosol to the membrane, the detailed motion of membrane ruffling was not clearly resolved in the volumetric images. To observe the detailed motion during membrane ruffling, we studied the distribution of filamentous actin in the cells expressing F-tractin-mCherry-p2a-CRY2-mCFP-iSH-p2a-CIBN-EYFP-CAAX. After stimulation (white cross in Fig. 4a region 1), the membrane ruffled around the area under stimulation and moved inward at the speed of ~1 μm/sec where the moving distance of the membrane edge along the white arrow is plotted in the inset. To demonstrate the capability of controlling the location of the stimulation area, the stimulation beam was moved to the newly formed membrane (white cross in Fig. 4a region 2) by shifting the Bessel beam, which was achieved by changing the SLM pattern with the X-shift as shown in Fig. 1b. The magnitude of membrane ruffling at region 2 was larger than at region 1 (detailed

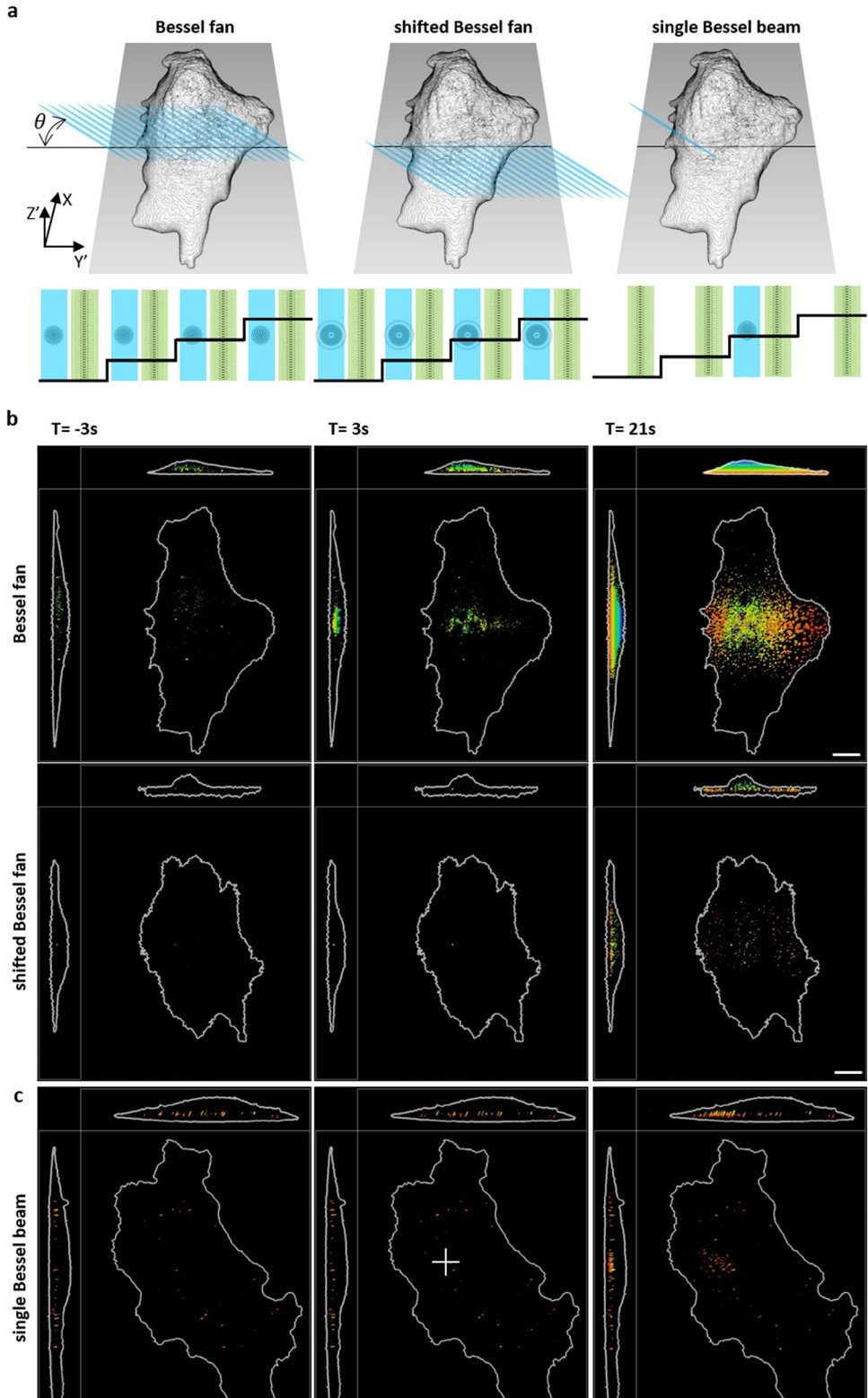

**Fig. 2 3D image recordings of CRY2-olig proteins at different activation schemes. a** The schematic for various activations: Bessel fan, shifted Bessel fan, and single Bessel beam. The activation beam illuminates on the Y'Z' plane at an angle of 32.8°. Bottom: The corresponding timing diagrams for the SLM patterns with laser 488 nm Bessel beam (blue) and 560 nm lattice beam (green) illumination synchronized with sample piezo waveforms (black). **b** The time-lapse images of optical induced clustering of CRY2olig-mRuby3 in XY' Y'Z' XZ' planes stimulated by Bessel fan and shifted Bessel fan activation. The depth of the oligomerization is color-coded, where the process of oligomerization occurs only at the basal membrane by the shifted Bessel beam activation. **c** The center part of the cell marked by the white cross is illuminated by the single Bessel beam. Oligomerization occurs near the activation site. Scale bar 10 μm.

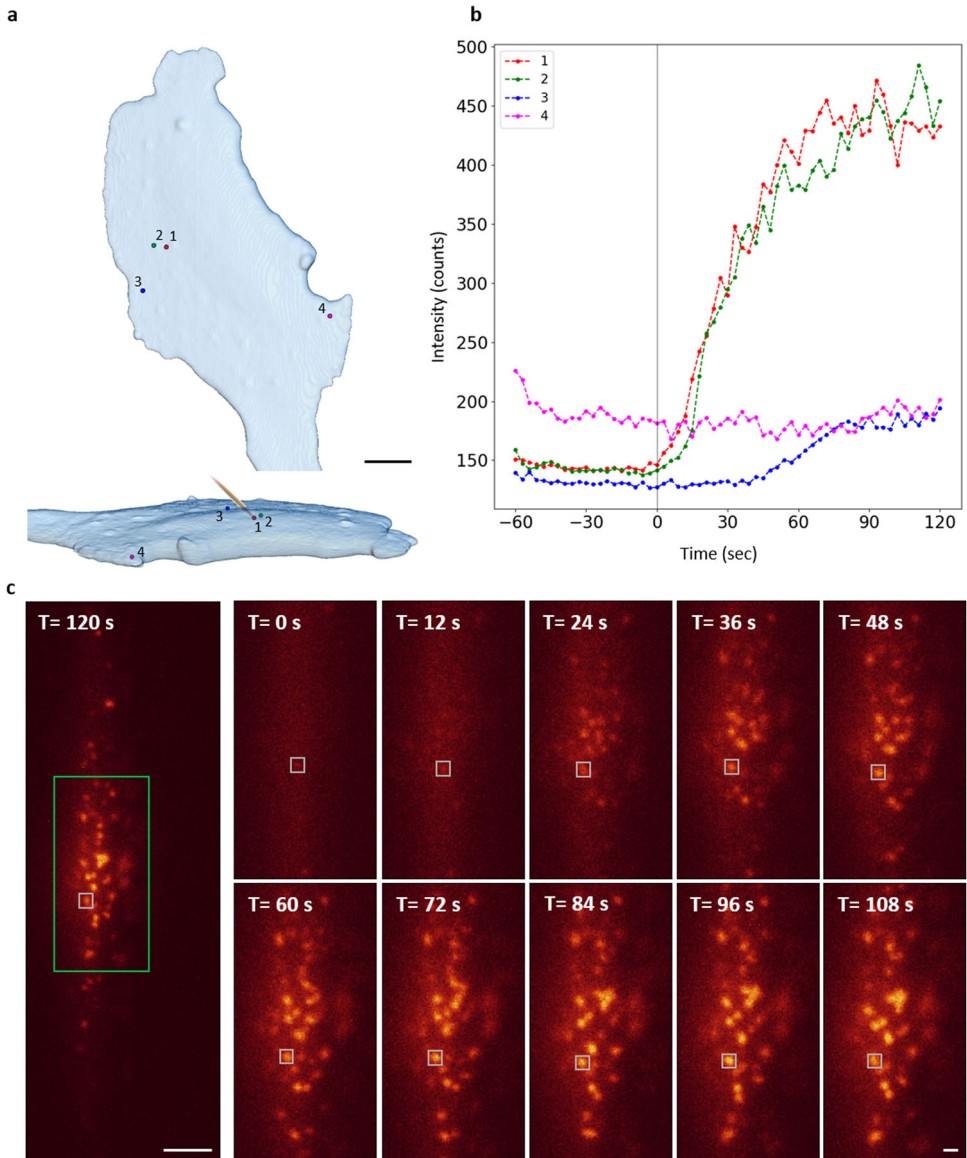

**Fig. 3 Time laspses images of the CRY2olig clusters activated by a single Bessel beam. a** Top view and side view of the 3D cell volume rendering image of the selected clusters of CRY2olig-mRuby3 molecules induced by a single Bessel beam activation at the spot shown in Fig. 2c. Scale bar 10 μm. **b** Time-lapse fluorescence intensity plots of the marked clusters in **a**. **c** Left: Raw image of the optically induced clustering of CRY2olig-mRuby3 in the selected plane activated by a Bessel beam marked in a white cross in Fig. 2c at T = 120 s. Right: Enlarged raw time-lapsed images of the green box in the left in the single Bessel beam activation experiment. Scale bar 10 μm and 1 μm, respectively.

membrane ruffling can be found in Supplementary Movie 7). To present the dynamics of the filament actin, we used color-coded signals at different time points, as shown in Fig. 4a. To better illustrate the dynamic behavior of the filamentous actins (the enlarged part of Fig. 4a), the contour function of the Amira software was used to outline the shape of the cell using the fluorescent signal from F-tractin-mCherry. The membrane protrusion and retraction upon the activation, as shown in Fig. S8, reveals that the membrane ruffling is presumably increased at the site of activation as compared to the rest of the cell body. However, after the second stimulation at the newly formed protrusion (region 2), the membrane ruffled rapidly in 3D at the edge of the cell as seen by the projection images in XY' and XZ of long-range actin movements (Fig. 4a). These results demonstrate that our system is capable of monitoring and activating the optogenetic molecules at the subcellular level.

**Long-term manipulation of cell migration by optoLLSM.** To validate the capability of our technique with respect to manipulating cellular behavior for an extended period, we first established an optogenetic system to control the migration of a cell in which the epidermal growth factor receptor (EGFR) was fused with CRY2olig and three repeating mApple (EGFR-CRY2olig-mApplex3) to activate the intracellular signaling of EGFR through blue-light induced clustering[32, 33]. By maximizing the brightness of red fluorescence protein for whole-cell imaging, the cross-talk activation could be minimized at a very low laser power. EGFR activation is known to trigger cell migration through the PI3K signaling[34, 35]. When the cells expressing EGFR-CRY2olig-mApplex3 were activated by the Bessel fan in the trailing edge, the formation of lamellipodia in the trailing edge was observed, as shown in Fig. 4b. The distribution of EGFR is color-coded at different time points. From Fig. 4b and Supplementary Movie 8, it can be seen that the direction of migration

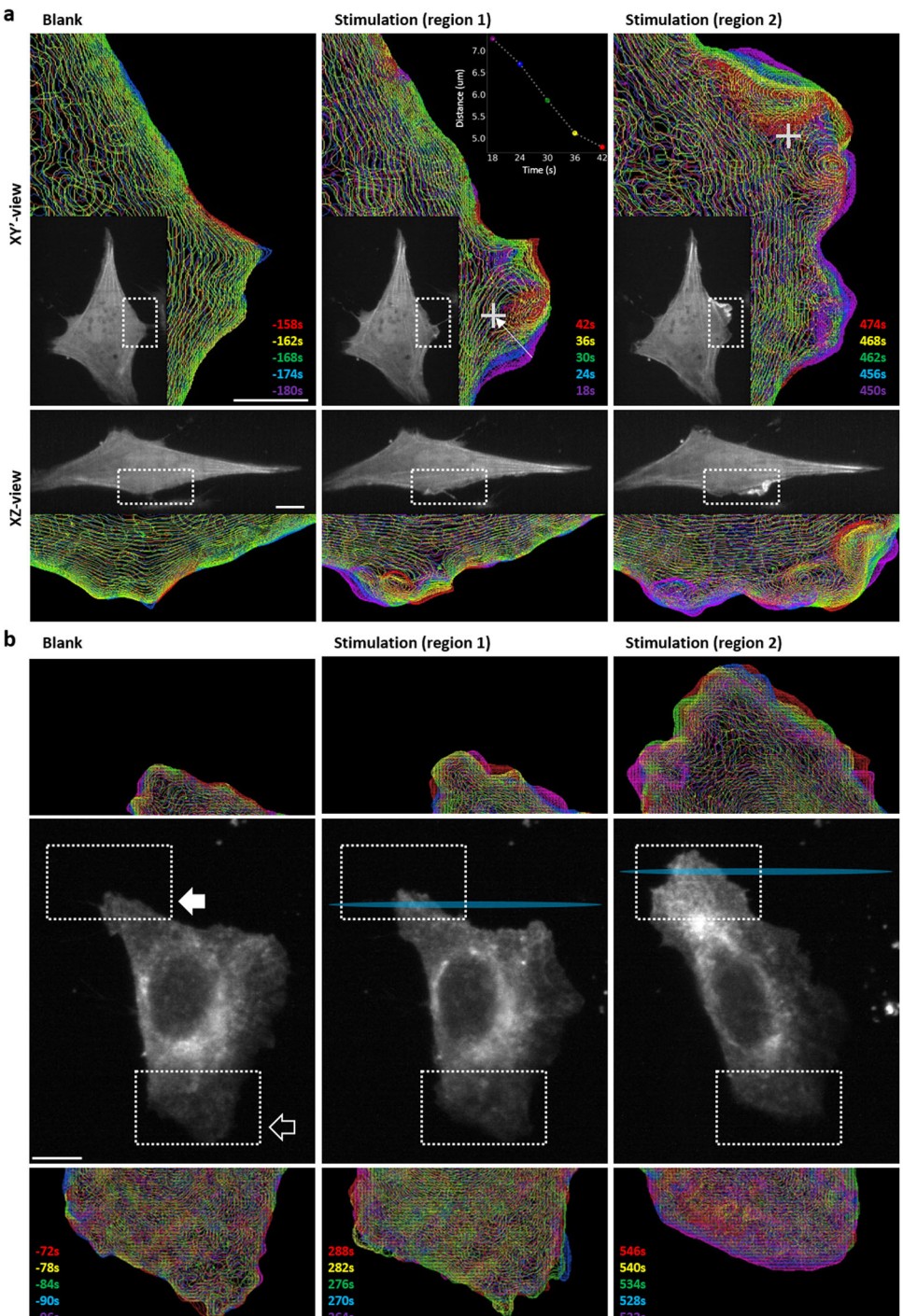

**Fig. 4 The dynamics of the cellular membranes upon sub-cellular activation. a** The cells expressing F-tractin-mCherry-p2a-CRY2mCFPiSH-p2a-CIBNEYFP-CAAX are illuminated by a single Bessel beam activation. The cross represents the activation site. To demonstrate the dynamic behavior of filamentous actin, intensity contours were plotted before and after the stimulation. At region 1, the distance of the membrane edge along the white arrow is shown in the inset. Different color represents different time points after stimulation. The top images are XY orthographic view, and the bottom images are XZ view with 32.8° rotation on X-axis. The gray images are the original images. **b** The Bessel fan was used to stimulate the endothelial growth factor receptor (EGFR)-CRY2olig-mApplex3-expressing cells at the trailing edge. The images in the center row are the whole-cell images at different times post-stimulation. The stimulation areas are marked as cyan. The solid arrow indicates the original trailing edge, and the hollow arrow indicates the original leading edge. Images on the top and bottom rows are enlarged views of the dashed box area in the middle row. Different colors represent different time points. Scale bar 10 μm.

was altered after stimulation in which the formation of lamellipodia was observed in the original trailing edge, and the old leading edge had started to retract. These results confirmed that the direction of migration could be controlled by optically induced clustering of the EGFR-CRY2olig-mApplex3 system activated by the Bessel fan at 488 nm in optoLLSM.

Next, a directed migration experiment was conducted by continuous stimulation. Since the cell would migrate out of the

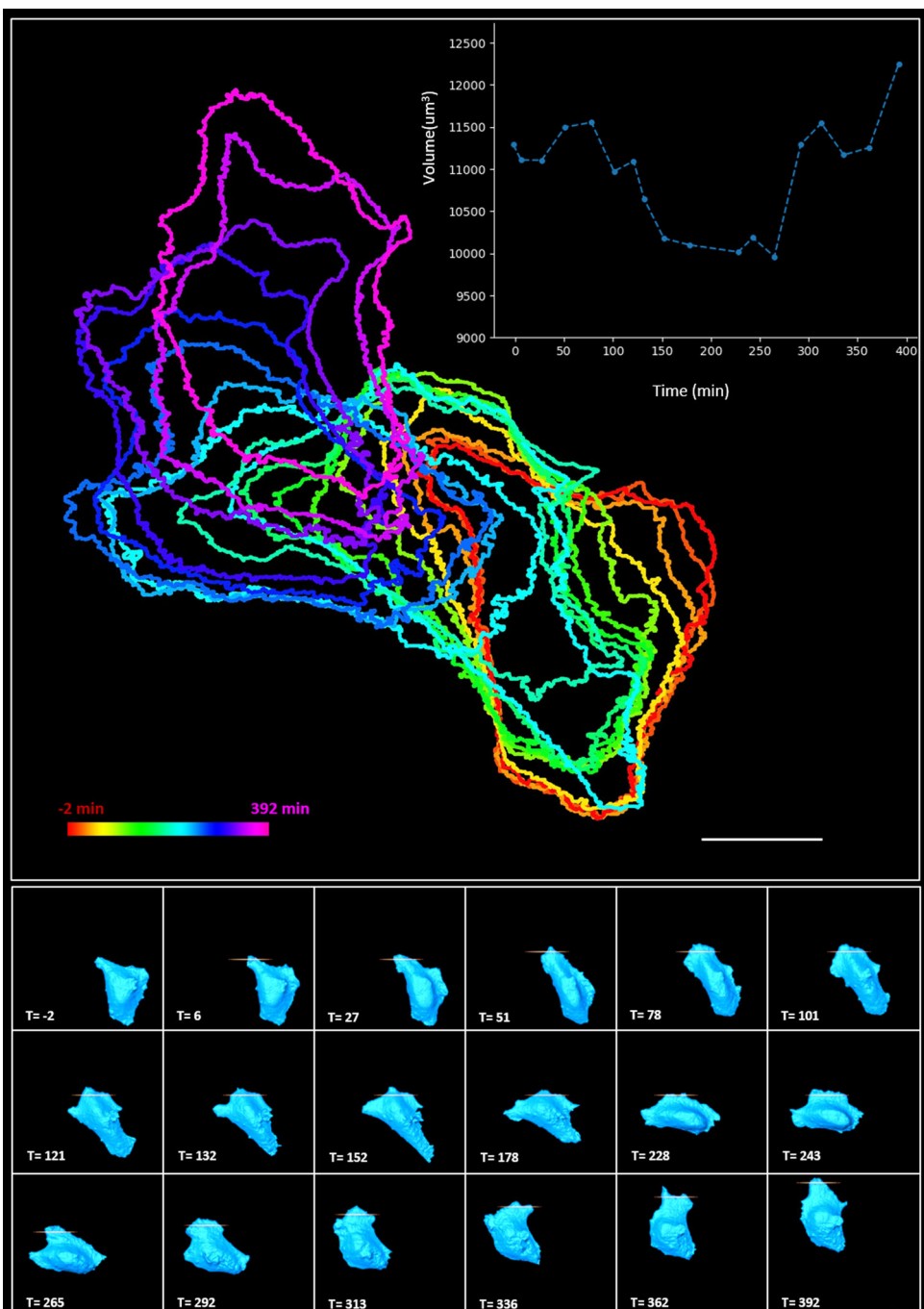

**Fig. 5 The time-dependent trajectories of the cell in the guided migration experiment.** The cell expressing EGFR-CRY2olig-mApplex3 was stimulated by a 1 nW Bessel fan. Different colors represent different time points. The individual LLSM 3D rendering images of the cell at different times, and the stimulated areas are depicted at the bottom. The volume changes of the cell during the cell-guided migration are plotted in the inset. The interval between each stimulation and image is about 1 min. The entire experiment lasted 392 min. Scale bar 20 μm.

field of view, the position of the samples every 20–30 min was adjusted while maintaining the same stimulated area in the cell, as shown in Supplementary Movie 9. To illustrate the trajectories of the cell in the same image, we marked the boundary of the cell at each time point with different colors. The time-dependent trajectories of the cell expressing EGFR-CRY2olig-mApplex3 stimulated by a Bessel fan in optoLLSM are shown in Fig. 5. The individual LLSM rendering images of the 3D cell and the stimulated area are depicted at the bottom of Fig. 5 and Supplementary Movie 10. Moreover, the volumes of the migrating cell over time are calculated and plotted in the inset, where the volume of the cell reduces when migration direction changes at T = 265, followed by the formation of large lamellipodia at T = 292. From these results, we demonstrated that the optoLLSM with Bessel fan stimulation was useful for continuously manipulating and monitoring EGFR-CRY2olig-mApplex3-expressing cells for more than 6 h, where ~75,000 2D-images (or 463 volumetric images) were collected, a finding that introduces the opportunity for conducting long-term optogenetic experiments in living cells.

## Discussion

In optoLLSM experiments, an additional stimulating light source is used. Therefore, we would like to evaluate the energy level used in the activation process. The minimum energy required for activation of the Bessel beam at different wavelengths was measured. It was found that the minimum energy required to induce the clustering of CRY2olig-mRuby3 molecules was 0.2 nW at 488 nm, and 1 nW was enough to activate the CRY2olig-mRuby3 system with a Bessel beam at 405, 445, 488, and 514 nm. For volumetric imaging in LLSM, the total power was 40 nW at 561 nm, which was distributed over the 145 lattice beams for the collection of fluorescent images with a signal-to-noise ratio greater than 5 for the observations of CRY2, CRY2olig, filament-actin, and EGFR.

The lattice lightsheet is produced by constructive interference of several non-diffracting Bessel beams with a minimized energy distribution in the sidelobes. However, when a single Bessel beam is used as the activation source, the influence of the energy distribution in the sidelobes, which could deteriorate the spatial resolution for activation, should be considered. A detailed comparison between the Bessel beam and Gaussian beam for lightsheet microscopy can be found in the Supplementary Methods, Figs. S9 and S10. The optical scheme for the optoLLSM is shown in Fig. S11. In this experiment, the spatial distribution of the optogenetic clustering of CRY2olig-mRuby3 using different Bessel beams formed by various minimum NAs with the same maximum NA was evaluated. The energy distributions of the Bessel beams at the back aperture and focal planes (XZ) can be found in the first and second rows of Fig. S3a of the Supplementary Information.

The LLSM images of the optogenetic clustering in U2OS cells expressing CRY2olig-mRuby3 stimulated by other Bessel fans at different time points are found in the fifth to the eighth row. Since the raw LLSM images contained all signals from CRY2olig-mRuby3, the Threshold function in the Amira software (Thermo Fisher Scientific) was selected to calculate the spatial distribution of the optogenetic clustering (an example is shown in Fig. S3b). The measured width of the Bessel fan and the width of optogenetic clustering are depicted in Fig. S3c. It was observed that the spatial distributions of the optogenetic clustering were much wider than the width of the stimulating Bessel beams, with ratios ranging from 4 to 5. The measured Bessel beam profiles in dye solution using different numerical apertures (NAs), as shown in Fig. S3, also confirmed the agreement between the simulations and the experimental data. To characterize the Bessel fan and single Bessel beam activation, we measured the fluorescence signals of the photoactivated CRY2olig-mRuby3 molecules by 488 nm in Fig. S12 (Supplementary Movies 11 and 12). The weak fluoresce signals excited by 488 nm can be used to visualize the activated CRY2olig-mRuby3 molecules within the path of the photoactivation beam, which could be used to indicate the beam profile of the photoactivation beam. In the Bessel fan activation, a strip with 2 um width (FWHM) was observed from the contributions of the main and side lobes of the illuminated Bessel beam, whereas a weak measured photoactivation was observed at the selected z slice in the single Bessel beam activation.

In the optogenetic heterodimer system (CRY2-CIBN), two independent membrane ruffling responses with the separation of 15 um could be triggered in the same cell with 1 nW Bessel fan activation at 488 nm (Fig. S7), which clearly demonstrates our capability for manipulating the cells with subcellular resolution. It was also found that about 30 min was required for the cell to recover to its original state at 1 nW stimulation. The energy used for CRY2olig-mRuby3 activation was similar to the CRY2-CIBN dimerization energy. To control cell migration, CRY2olig was linked to the intracellular part of the EGFR, in which optically-induced oligomerization of CRY2olig led to EGFR oligomerization. It is known that the existence of other EGFR molecules in close proximity to each other will promote EGFR signal activation. Therefore, the direction of migration could be reversed by activating EGFRs in the trailing end of the migrating cells, as shown in Figs. 4b and 5. Note that due to the individual cultured cells with different transfection efficiency, morphologies, and adherence to the coverslip, the quantified measurements in this study are based on the observations from indivudal example.

To achieve cell manipulation, a focus shift was used in this experiment in which regions illuminated by the lightsheet could be controlled by adding a binary phase Fresnel lens on the spatial light modulator used to generate Bessel beams filtered by a passive optical component, physical mask. As shown in Fig. 1b, the region illuminated by the lightsheet, whose length and thickness are pre-determined by a chosen annual mask, could be moved forward or backward in the light propagation direction. Within the length of the lightsheet, the optogenetic system could be activated effectively, whereas the energy distribution of the lightsheet diverged rapidly beyond the length of the lightsheet but was not enough for optogenetic system activation (energy distribution is shown in Fig. S2). Therefore, the optogenetic molecules located on the membranes could be activated locally at the apical or basal membranes to achieve a controllable optogenetic system in subcellular fractions. Despite the contribution from the sidelobes of a Bessel stimulation, which can be minimized by carefully engineering the beam shape before the experiment as shown in Fig. S12, not only the diffusion effect of CRY2olig-mRuby3 but also the energy distribution in the sidelobes still contributed to the broadened width in the activated area as shown in Fig. S3c. To improve the spatial confinement for activation, two-photon activation or temporal focusing may be used[36, 37]. However, the integration of these techniques into SLM based LLSM may be complicated, especially for altering the activation positions programmatically with the present acquisition software. Another challenge in implementing 3D activation in LLSM was the line confinement activation instead of a point in the propagation direction, which was about 20 μm at the present configuration in order to cover the whole cell imaging. Such confinement allows us to conduct local activation of molecules located on the apical or basal membrane by SLM manipulation but not enough to activate molecules within a specific organelle in the cells with the fixed annual mask. An aperture-free technique for the generation of activation beam and lattice beam at different beam lengths is required in the future. In summary, the subcellular activation capability of optoLLSM opens the opportunity for optogenetic activation of single cells in small animal models.

## Materials and methods

**Plasmid construction.** For the multiple-gene expression system, the plasmid pCR3.1-p2a-SpeI was generated from the original pCR3.1 plasmid, which was added in-frame to the porcine teschovirus-1 2 A self-cleaving peptide (p2a) and SpeI cutting site to fit with the NheI cutting site of the enhanced green fluorescent protein (EGFP) series of vectors. The F-tractin-mCherry-p2a-SpeI, CRY2-mCherry-iSH-p2a-SpeI and CRY2-mCFP-iSH-p2a-SpeI were constructed based on pCR3.1-p2a-SpeI plasmid. The filamentous actin marker (a cytoplasmic actin filament reporter, the neuronal inositol 1,4,5-triphosphate 3-kinase A actin-binding domain known as F-tractin [Plasmid #58473, addgene]) was fused with a monomeric red fluorescent protein to the N-terminal p2a peptide via digestion using two restriction enzymes, NheI-HF and PmlI, and ligated into pCR3.1-p2a-SpeI plasmid. Using the specific primer polymerase chain reaction (PCR) cloning, the C-terminal of Arabidopsis cryptochrome 2 (CRY2) with mCherry or mCFP (monomer cyan fluorescent protein) was linked with the phosphoinositide 3 kinase (PI3K)-catalyzed motif (iSH) to generate CRY2-mCherry-iSH-p2a-SpeI and CRY2-mCFP-iSH-p2a-SpeI plasmids. The multiple gene expression of the backbone plasmid, which contains the CRY2 interaction binding protein N-truncated version (CIBN) domain linked to the plasma membrane anchor leading sequence (CIBN-CAAX), was cut by double digestion using NdeI and NheI. The CRY2-mCherry-iSH-p2a-SpeI or CRY2-mCFP-iSH-p2a-SpeI was then cut by double digestion using NdeI

and SpeI for insertion into the backbone plasmid. At the end, the same protocol for generating CRY2-mCherry-iSH-p2a-CIBN-EYFP-CAAX and F-tractin-mCherry-p2a-CRY2-mCFP-iSH-p2a-CIBN-EYFP-CAAX plasmid was followed. On the other hand, the generation of CRY2olig-mRuby3. CRY2olig-mCherry plasmids were modified by point mutation PCR to generate an XmaI cutting site for compatible cohesiveness with the AgeI cutting site. The CRY2olig-XmaI-mCherry plasmids were then digested with XmaI and BsrGI. Using an AgeI and BsrGI double digestion, the mRuby3 was inserted into the CRY2olig-XmaI plasmid. To prepare CRY2olig-mApplex3, the BsrGI-mApple-BsiWI-PmlI was first created by specific primer PCR to increase the repeat of mApple, and AgeI and BsiWI (compatible cohesive end with BsrGI) were used to replace the original mRuby3 site of CRY2olig-mRuby3. The XhoI cutting site of EGFR fits in the frame with the open reading frame of CRY2olig-mApplex3. EGFR was cut with NheI and XhoI for insertion into the N-terminus of CRY2olig-mApplex3 for creating EGFR-CRY2olig-mApplex3. All aforementioned plasmids are available upon request.

**Imaging of optogenetic tools expressed cell line**. U2OS cells were maintained in Dulbecco's modified Eagle's medium (DMEM, Gibco, ThermoFisher), which contained 10% fetal bovine serum (FBS) from Gibco (ThermoFisher), 2 mM L-glutamine (Gibco, ThermoFisher), and 1% penicillin-streptomycin solution (Gibco, ThermoFisher). Before transfection, 5 mm glass coverslips were placed in 1X phosphate-buffered saline (PBS) buffer with 10 μg fibronectin for 2 h at room temperature. The fibronectin-coated coverslips were washed twice with 5X PBS and once with 1X PBS. U2OS cells ($1 \times 10^5$) were seeded into 12-well plates containing the fibronectin-coated glass coverslips and cultured in DMEM medium without penicillin-streptomycin to enhance transfection efficiency.

Cells were incubated for 16 h at 37 °C and 5% $CO_2$ before transfection with 0.8 μg plasmid DNA per well using Maestrofectin Transfection Reagent (Omics Bio). After 16 h, the transfected cells were allowed to recover for 24 h in 1% penicillin-streptomycin and 10% FBS DMEM medium.

**Stimulation setting and image acquisition**. To standardize the stimulation conditions, we used a power meter (PD300-1W with Nova II, Ophir Photonics, USA) to detect the input power of the Bessel beam behind the pupil of the exciting objective. The required parameters for different wavelengths (405, 445, 488, and 514 nm) and power (0.5, 1, 2, and 4 nW) were adjusted according to the readings. The detection limit of the laser power meter is 1 nW. To obtain laser power of 0.5 nW, the laser power was first set at 5 nW according to the power meter and then attenuated linearly by a combination of neutral density filter and acousto-optic tunable filter (AOTF). Multi-color scanning was used in the Z-scan mode in the SPIM program (licensed by Howard Hughes Medical Institute, Janelia Research Campus) to generate multi-color 3D images, on which the signals from each channel were collected one by one from the same layer. As shown at the bottom of Fig. 2a, the SLM displaying two pre-loaded color patterns (488 nm and 560 nm) for either Bessel/lattice beam or shifted Bessel/lattice beam at each step of the sample stage at the designed time sequence based on the experimental exposure time in the SPIM software. In an optogenetic experiment, the excitation source was first switched to the projected Bessel beam mode for stimulation followed by lattice beams for imaging. In each experiment, blank control images were collected for the first 20 time points by turning off the stimulating light source with a mechanical shutter which is placed in front of the 488 nm laser, controlled by a homemade Arduino device (Arduino Uno) and Labview program to synchronize the image acquisition software, SPIM. For single Bessel beam stimulation, the SPIM program held the x-galvanometer without any movement, thus allowing the Bessel beam to stimulate the central area of the cells, and the activation wavelength 488 nm was illuminated on the samples at a selected z slice while performing z-stacking; the timing is clocked by Arduino device.

In the experiment requiring moving the stimulation area along the X-axial direction, the position of the Bessel pattern on the SLM could be changed to illuminate the corresponding focal plane position. Alternatively, the x-galvo could be used to move the illumination area of the Bessel beam to a specified position with an accuracy of 0.1 um. In the single-Bessel beam stimulation experiment, a shutter was used to allow the stimulation beam to activate the cells once per volume scan. In addition, the tiling lattice lightsheet technique was used in the experiment to shift the Bessel beam along the Y-axial direction by changing the beam shift coefficient as described previously[27].

**Image capturing and processing**. The details for the optoLLSM optical scheme can be found in the Supplementary Methods and Fig. S11. In this experiment, a 2048 × 2048 pixel COMS camera (ORCA Flash 4.0 V2, Hamamatsu) was used in which an area of 200 × 960 pixels (20.6 μm × 98.9 μm) was used for LLSM imaging. In a typical LLSM imaging setup, the exposure time for each frame is 9 ms, the sample scanning step is 0.6 μm, and 131 planes of images are used for volumetric imaging. For time-lapse volumetric imaging, the volumetric image is recorded every 3 s. The camera settings for the directed cell migration experiment are the same (10 ms exposure time for 200 × 1024 pixels with 161 planes) for both colors 488 nm (activation) and 560 nm (imaging), where it takes ~3 s to finish 3D optogenetic imaging. Initially, the time interval between each volumetric image was set to 6 s until we observed cellular response or membrane

ruffling (in this case, 60-time points). We then changed the time interval to 1 min to observe directed cell migration for 343-time points. Note that for cell manipulation, such as induced membrane ruffling and directed cell migration experiments, the subcellular responses of cells cannot be predicted. Therefore, we cannot program the SLM pattern time profile in advance. To demonstrate the capability of local activation, the photoactivation was conducted manually. After recording the first cellular response, the program was stopped, and the activation beam was moved to the desired area either by changing the SLM pattern or the x-galvo position. The program was resumed to record the cellular response. These processes were repeated until the end of the experiment. Since the imaging plane was tilted from the normal direction of the coverslip, the images orthogonal to the coverslip with a view angle similar to conventional microscopy were reconstructed from the raw images through the GPU program. To process the images for better presentation in Figs. 4 and 5, Amira imaging process software (ThermoFisher) was used. For the directed cell migration, the timing and position were registered in the movie reconstruction, as shown in Fig. 5 and Supplementary Movie 10.

**Reporting summary**. Further information on research design is available in the Nature Research Reporting Summary linked to this article.

## Data availability
The source data behind the graph in Fig. 3b can be found in [https://doi.org/10.6084/m9.figshare.20382468.v1]. Representative movies for each figure are provided as Supplementary Movies 1–12. All other data are available from the corresponding author on reasonable request.

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

## Acknowledgements

The microscope control software is licensed by the Howard Hughes Medical Institute, Janelia Research Campus. This work was supported by the Ministry of Science and Technology, Taiwan (Project No. MOST 109-2113-M-001 -034 -MY3 to P.C., 109-2628-M-001-001-MY4, 110-2321-B-002-012 to B.C.C., MOST 110-2636-B-007-011, and MOST 111-2636-B-007-009 to Y.C.L.). This work was also supported by the Academia Sinica, Taipei, Taiwan (AS-IA-110-M04 to P.C and AS-CDA-107-M08 to B.C.C.), and National Tsing Hua University, Hsinchu, Taiwan (111Q2713E1 to Y.C.L.). We also thank the DNA Sequencing Core Facility of the Institute of Biomedical Sciences, Academia Sinica, for DNA sequencing analysis. The core facility is funded by Academia Sinica Core Facility and Innovative Instrument Project (AS-CFII-108-115).

## Author contributions

W.C.T. and B.C.C. planned and performed the imaging experiments and image processing. W.C.T.,Y.T.L, C.H.L. and B.C.C. designed and constructed the optoLLSM. W.C.T., C.H.Y., S.W.C, and L.G. performed imaging analysis and Bessel beam characterizations. W.C.T., and C.H.T. performed experiments on plasmid construction and transfection. Y.L.L., Y.C.L., and T.L.H. provided the bio-techniques about the construction of the plasmids and cell lines. B.C.C. and P.C. wrote the manuscript and managed the project.

## Competing interests

The authors declare no competing interests.
