## [Peer Review File · Communications Biology]

Reviewers' comments:

Reviewer #1 (Remarks to the Author):

In this manuscript, "Long-term three-dimensional optogenetic manipulation of cell migration using lattice lightsheet microscopy" Tang et al. demonstrate the combination of Bessel-beam based photoactivation together with lattice light sheet microscopy. By shifting the Bessel beam image on the SLM, photoactivation can be positioned in three-dimensions. The paper is appropriate as a technical demonstration and may be suitable for publication in Communications Biology. However, I'm not entirely convinced that true "three-dimensional" photoactivation can be achieved with the current setup. A more detailed characterization of the x,y, and z resolution of the photoactivation would be very helpful. More specific comments are described below.

Specific comments:

1) My primary concern is related to the claim of "precise activation in 3D", which I don't believe is convincingly demonstrated. More detailed characterization of the 3D x, y, z resolution of the photoactivated region would be helpful. It seems that using an elongated Bessel beam for photoactivation would actually give substantially worse resolution along the beam propagation axis than a confined Gaussian beam. A more clear description of why a Bessel beam was chosen and how the beam parameters affect the photoactivation resolution would be helpful.

2) Also, I'm unclear on the motivation for the maximum and minimum NA's of the Bessel beams chosen for photoactivation. Is there is a cutoff for minimum local intensity at the sample beyond which photoactivation will occur. How do the relative intensities of the main vs. side lobes of the Bessel beams factor into the lateral resolution estimates? Furthermore, I'm unclear how the resolution estimate of 20 microns along "y" was determined. Bessel beams (and Gaussian beams) can be generated with a much shorter beam length. Wouldn't these provide better resolution along the propagation axis?

3) It would be very helpful if the authors could provide power estimates for imaging and photoactivation at the sample plane (e.g. W/cm²) rather than just total power at the rear pupil of the excitation objective. Presumably the photoactivation efficiency is sensitive to the instantaneous local intensity rather than the total delivered dose.

4) The authors do demonstrate some enrichment in photoactivation at either the apical or basal cellular membrane in Figure 2 by shifting the Bessel beam in "y" using a defocused pattern on the SLM, however, this isn't really "precise 3D activation" as it's achieved by shifting a rather long activation beam beyond the physical extent of the sample rather than by confining it to a specific cellular region. Moreover, no quantification of the relative activation at the apical or basal membranes is provided and Figure 2 appears to show only a very minor difference between the "Bessel fan" and "shifted Bessel fan" conditions. Quantifying the axial activated area (similar to as was done for figure S2) would be helpful.

Minor comments:

1) Pg 7 "Each imaging volume consisted of 131 stacks of images, which took 2.6s to collect". I'm assuming this is meant to say "131 2D images" rather than "131 stacks of images".

2) "RFP" is used as a general abbreviation for "red fluorescent protein" which is then followed by a specific description of the fluorophore-conjugate (e.g. F-tractin-mCherry). This is confusing as RFP is generally an abbreviation for the specific red fluorescent protein (e.g. mRFP). It would be better just to list the specific proteins used.

Reviewer #2 (Remarks to the Author):

Major claims of the paper:

In this paper, the author claim to produce controled and local (in 3D) activation of optogenetic tools. They shows that local recrutement to the plasma membrane of a iSH domain(activator of

Cdc42/Rac1) promotes membrane ruffling and the the local formation of EGFR clusters promotes cell migration.

Novelty of the claims:

To my reading, this paper is meant to promote the use of LLSM to do local and long term optogenetic experiment. Then, the two biological experiments seem to be present just as a validation of the use of this microscopy setup.

I have a few comments about this novelty:

First of all, and not mentioned/referenced in the main text, researchers are now doing optogenetic experiment, with subcellular (~2-4um precision) control of activation, for more than 10 years, without the need of LLSM. They indeed used a wide range of setup from spinning disc+ frap modules, widefield microscope +dmd illumination setups, to scanning laser confocal microscopes+ frap routines, and most of the time managed to do experiment ranging over 12h experiments. For even better resolution, people already used some 2 photon microscopes coupled to optogenetic experiments and/or used optogenetic systems enabling the control of inactivation. This has to be mentioned.

Now concerning this article.

First, the LLSM setup and the possibility to do local illumination has already been described in reference 23.

Second, the author claims that they do local optogenetic activation, however, due to the diffusion properties of the optogenetic tool, the "local" appears on most of there movie "almost global". This could be improved by better quantification (more precise, and with statistics on more cells).

Third, the 2 validation experiments are interesting. However, the "local" activation ends up to be very broad. Then, it might be better to do a deeper characterisation of the processes involved here which are so far very descriptive.

Overall, I find that the title is an overstatement on many points and that, presented this way, this article is just a small technical adaptation of already existing/published tools.

Validity of the results:

Concerning the validity of the result, I do trust the implementation of a Bessel beam. However, concerning biological experiments, we do not have access to how reproducible are the experiments.

Moreover, there is no clear quantification of how spread is the activation, what are the different time courses, ...

Writing:

Last but not least, many parts of this article, starting with the abstract, should be rewritten to improve clarity.

Finally, figures are often missing a lot of information of interest (as for the scheme fig1a) and there captions should be improved.

All this together, I do not recommend this article for publication

Reviewer #3 (Remarks to the Author):

See attached document.

Summary:

Tang *et al.* describes a method of performing sub-cellular photoactivation in conjunction with lattice light-sheet microscopy (LLSM). Most notably, the authors perform these experiments without use of any additional modules or alterations to the original LLSM design through alterations of the spatial light modulator (SLM) patterns. To test their method, the authors developed two probes: (1) CRY2olig (optically induced clustering) and (2) CRY2-CIBN (triggering of regional membrane ruffling). Optically induced clustering experiments were used to demonstrate “Bessel Fan” photoactivation both throughout the entirety of the cell body and at the cell-substrate interface, as well as single Bessel beam photoactivation. Triggering of regional membrane ruffling was used to showcase the method’s ability to dynamically change the site of activation in short-term and long-term experiments. The authors clearly demonstrate that they are able to perform sub-cellular photoactivation using only the original LLSM design, which could be of great interest to the microscopy and biology communities. However, I have further questions regarding the implementation methodology, photoactivation precision, and image quantification.

Major Comments:

1. Given that movement of the beam position through alterations in the SLM pattern (doi:10.1364/OE.27.001497, cited in this work) and photoablation/FRAP/photoactivation with LLSM (<https://doi.org/10.1083/jcb.201906111>, <https://doi.org/10.1101/2020.09.01.276824>, <https://doi.org/10.1117/12.2509467>) have been previously shown, the novelty of this work is in the use of only the original LLSM design to perform sub-cellular photoactivation. This is important because it dramatically improves the ease of implementation and accessibility of the method for other groups with access to LLSM. However, this makes accurate reporting of the method of highest importance, and I feel as though the manuscript is currently lacking in sufficient detail. Specifically:
 - a. For single Bessel beam activation, how is illumination for a single slice practically achieved? It appears as though the activation is performed

during a z-stack, so how is the light gated for the rest of the acquisition?
The authors make reference to a mechanical shutter, but it is not evident how this is controlled dynamically during an experiment.

- b. For experiments where the activation beam position is changed during the experiment, how is this performed? It is evident that the position change is done by changing the SLM pattern, but is this done dynamically or does the experiment need to be paused to upload a new pattern to the SLM?
 - c. In the directed migration experiment, the authors reference the need to reposition the sample every 20-30 minutes. Does this imply that the acquisition was paused to move the sample and then restarted? How is the precise acquisition timing and sample position monitored during the experiment?
 - d. Overall, the heart of this paper seems to lie in the description of how the authors achieve sub-cellular photoactivation in the conventional LLSM system, for which the description is minimal. I would recommend for each specific variant (Bessel Fan, Y-Shifted Bessel Fan, Static Single Bessel, Dynamic Single Bessel), the authors should allocate a describe in detail how each of these experiments were practically realized. This is necessary to ensure reproducibility, and at present these experiments would not be reproducible by a group with access to LLSM.
2. One major claim repeated throughout this paper is that the necessary photoactivation power is on the order of a single nW. Can the authors provide a more in-depth description of how these measurements were performed? The original LLSM paper used excitation powers on the order single μ Ws at the lowest, and this work claims to perform volumetric imaging at 40 nWs. A 100 fold change in power seems highly unusual, and the power range of the power meter referenced ends at 0.5 nW, but the authors reference using powers of 0.2 nW which in principle cannot be determined with the referenced instrument. Can the authors please clarify exactly how these measurements were carried out and validated?

3. Any microscopy development is subject to a tradeoffs, and it is important to discuss these tradeoffs to help a reader weigh the pros and cons of implementing a technique. The authors have done well to highlight many of the strengths of their methods, and have begun discussing limitations at the end of the discussion section, but I think it would be beneficial to expand this discussion. For example, the annular mask cannot be dynamically switched during an experiment in the conventional LLSM system. Therefore, the inner and outer NA of the Bessel beam used for photoactivation is predetermined and the length of the beam will be similar to that of the lattice used for imaging. This means that the localization of photoactivation in the direction of propagation is somewhat limited by the depth of field needed for image acquisition. An expanded discussion of this limitation and others, contextualized with the benefits of the method, is necessary.
4. It is difficult to assess the specificity of the photoactivation given the diffusion of CRY2oligo-mRuby3 (as noted by the authors) and the nature of membrane ruffling itself. Because this paper fundamentally relies on sub-cellular photoactivation, I think it is important to have an accurate description and quantification of the area being activated. To assess this, I would suggest the authors perform a photobleaching assay on a fixed sample using each variant of photoactivation approach (Bessel Fan, Y-shifted Bessel Fan, Static Single Bessel, Dynamic Single Bessel) described in this paper (though the light intensity may need to be modulated to achieve significant photobleaching). This would allow the authors to show precision and accuracy of the photoactivation approach described in this manuscript.
5. Many of the conclusions of this study appear to be derived from representative examples as opposed to through quantification over number experiments. Specifically, the following conclusions are drawn without quantification of the associate images:
 - a. The minimum power for photoactivation is 0.2 nW.
 - b. Shifted Bessel Fan excitation concentrates photoactivation at the cell-substrate interface.

- c. Membrane ruffling is increased at the site of activation as compared to the rest of the cell body
- d. Directed migration can be induced through localized photoactivation.

The data presented by the authors is consistent with the conclusions that they have drawn, however it is not sufficient to state conclusions from single examples. Further repeated experiments and quantification are necessary to show these statements as fact. Otherwise, the authors must emphasize that these are simply observations from single examples as opposed to conclusions from repeated studies.

Minor Comments:

1. The authors should make reference to previous literature performing photoactivation and/or photobleaching with LLSM noted above.
2. For the directed migration experiment, what are the associated imaging parameters (exposure time, number of steps, time between volumes). It is unclear if this is the same as previous experiments or not.
3. In the final paragraph of the discussion, the authors reference a challenge in implementing 3D activation is the axial resolution. I am not sure what the authors are referring to at this point, as the axial resolution is the minimum distance at which two objects can be distinguish in the axial direction. I'm not sure how this connects to the 20 μm referenced by the authors or is a limitation of the system.

I intend for the above comments to be purely constructive to help improve the overall quality of the manuscript. I hope the authors and their families stay healthy during this time.

Reviewer #1 (Remarks to the Author):

In this manuscript, “Long-term three-dimensional optogenetic manipulation of cell migration using lattice lightsheet microscopy,” Tang et al. demonstrate the combination of Bessel-beam-based photoactivation together with lattice light-sheet microscopy. By shifting the Bessel beam image on the SLM, photoactivation can be positioned in three dimensions. The paper is appropriate as a technical demonstration and may be suitable for publication in Communications Biology. However, I’m not entirely convinced that true “three-dimensional” photoactivation can be achieved with the current setup. A more detailed characterization of the x,y, and z resolution of the photoactivation would be very helpful. More specific comments are described below.

Response: We thank the reviewer’s comments. To demonstrate the spatiotemporal control of photoactivation, in the revised manuscript, we added a new figure as figure 3 showing the time-dependent fluorescence intensity of the photoactivated clusters in different regions of the cell activated by a single Bessel beam where the time-dependent fluorescence intensities of the activated CRY2oligo-mRuby3 molecules in different region behaved differently indicating the photoactivation was confined in a small region initially during the photoactivation. To describe the spatiotemporal behavior of different photoactivated clusters, the following sentences are added on page 10, lin 165 of the revised manuscript “To analyze the spatiotemporal behavior of the activated clusters, 3D cell volume renderings with selected CRY2oligo-mRuby3 clusters are shown in Figure 3a, where cluster 1 is located right at the activation region, clusters 2 and 3 are 2 μm and 5 μm away from the activation region, respectively. The time-dependent fluorescence intensity changes of the chosen clusters are plotted in Figure 3b, where cluster 1 shows an instantaneous increase upon the activation beam. The delayed responses are observed according to the distance from the activation spot, for example, 10 s for cluster 2 and 40 s for cluster 3. To evaluate the photobleaching effect in our system, we monitored the fluorescence signal of a pre-existing cluster (cluster 4), which was far away from the activation beam. No decrease in fluorescence intensity of cluster 4 was observed during the whole experimental period indicating that the photobleaching effect of our system is negligible. A series of raw images for the selected plane activated by a single Bessel beam are shown in Figure 3c, where the formation of the cluster over time is marked in a white box. The analysis of fluorescence intensities of different clusters activated by Bessel fan and shifted Bessel beam is shown in Figures S5 and S6. These findings clearly demonstrate that LLSM could be easily modified to conduct 3D optogenetic experiment with good spatial control using a regular and modified Bessel fan/beam as an activation source.” We also characterized the spatiotemporal behavior of the photoactivated clusters for three different photoactivation schemes. The time-dependent fluorescence intensities for different activation schemes can be found in the new Figures 3, S5, and S6. From these results, we demonstrated that the photoactivation could be initially confined in a local region. However, the diffusion of the photoactivated clusters smeared out the effect of local activation at a later time. Therefore, we agree with the reviewers’ concerns on the use of “precise photoactivation in 3D”. In the revised manuscript, “precise activation in 3D” was replaced by “better spatiotemporal control of photoactivation” in line 25 and “controllable activation in 3D” in line 87.

Specific comments:

1) My primary concern is related to the claim of “precise activation in 3D”, which I don’t believe is convincingly demonstrated. More detailed characterization of the 3D x, y, z resolution of the photoactivated region would be helpful. It seems that using an elongated Bessel beam for photoactivation would actually give substantially worse resolution along the beam propagation axis than a confined Gaussian beam. A more clear description of why a Bessel beam was chosen and how the beam parameters affect the photoactivation resolution would be helpful.

Response: As mentioned above, the profiles of Bessel fan and single Bessel beam activation were characterized by measuring the fluorescence signals of the photoactivated CRY2oligo-mRuby3 molecules at the photoactivation wavelength, 488 nm. The difference between the Bessel fan activation and shifted Bessel activation is the axial position (y in Figure 1 according to our definition) of the Bessel beam where the Bessel fan across the whole cell and the shifted Bessel beam only covered the apical or the basal membrane of the cell. The weak fluoresce signals excited by 488 nm can be used to visualize the photoactivation beam profile at the photoactivation plan. In the Bessel fan activation, a strip with a 2 um width (FWHM) was observed in the new movie 11, where the energy distribution of the Bessel beam was shown in the new Figure S12. On the other hand, a weak activation was observed for single Bessel activation at the selected plane, as shown in the new movie 12. Despite that the CRY2oligo-mRuby3 can be activated in a confined region, the diffusion of the photoactivated molecules may result in global activation instead of local activation. To evaluate how CRY2oligo-mRuby3 molecules responded to photoactivation both in space and time, we calculated the fluorescence intensity of photoactivated clusters in different parts of the cells. The time-dependent fluorescence intensity for three different types of activation schemes is plotted in the new Figures 3, S5, and S6. We added the data for single Bessel beam excitation in the revised manuscript as Figure 3 to illustrate the spatiotemporal control of the photoactivation. From figure 3, we can see that the CRY2oligo-mRuby3 molecules formed clusters (#1) upon the photoactivation at the activation zone, whose fluorescence intensity increased right after the activation. Near the activation region, clusters were formed at a delayed time. For example, the intensity of cluster #2 increased with 3-6 seconds delayed with respect to cluster 1, where cluster #2 was about 2 um away from cluster #1. At 5 um away from cluster #1, the fluorescence intensity of cluster #3 increased with a significant delay time (~ 60s), and the final intensity of cluster #3 was only a fraction of cluster #1. From these results, we can see that the initial activation was confined to a small region. However, the diffusion of the activated molecules and the pre-existing aggregate of CRY2oligo-mRuby3 molecules complicated the photoactivation process. In the new figure 3, we also monitored the fluorescence intensity of a pre-existing cluster (#4) located far away from the activation region. The fluorescence intensity of cluster #4 stayed the same during the whole observation period indicating that there was minimal photobleaching effect for the imaging process. Following sentences were added in the revised manuscript on page 10, lin 165 to describe the spatiotemporal behavior of these activated clusters “To analyze the spatiotemporal behavior of the activated clusters, 3D cell volume renderings with selected CRY2oligo-mRuby3 clusters are shown in Figure 3a, where cluster 1 is located right at the activation region, clusters 2 and 3 are 2 μm and 5 μm away from the activation region, respectively. The time-dependent

fluorescence intensity changes of the chosen clusters are plotted in Figure 3b, where cluster 1 shows an instantaneous increase upon the activation beam. The delayed responses are observed according to the distance from the activation spot, for example, 10 s for cluster 2 and 40 s for cluster 3. To evaluate the photobleaching effect in our system, we monitored the fluorescence signal of a pre-existing cluster (cluster 4), which was far away from the activation beam. No decrease in fluorescence intensity of cluster 4 was observed during the whole experimental period indicating that the photobleaching effect of our system is negligible. A series of raw images for the selected plane activated by a single Bessel beam are shown in Figure 3c, where the formation of the cluster over time is marked in a white box. The analysis of fluorescence intensities of different clusters activated by Bessel fan and shifted Bessel beam is shown in Figures S5 and S6. These findings clearly demonstrate that LLSM could be easily modified to conduct 3D optogenetic experiment with good spatial control using a regular and modified Bessel fan/beam as an activation source.” From these results, we demonstrated that the photoactivation could be initially confined in a local region. However, the diffusion of the photoactivated clusters smeared out the effect of local activation at a later time. We agree with the reviewers’ concerns on the use of “precise photoactivation in 3D”. In the revised manuscript, “precise activation in 3D” was replaced by “better spatiotemporal control of photoactivation” in line 25 and “controllable activation in 3D” in line 87.

As for the comment that the elongated Bessel beam would provide worse resolution in photoactivation compared with the Gaussian beam, we apologize for the confusion. In lightsheet microscopy, the imaging plane is perpendicular to the excitation plane, as shown in Figure 1a, Figure S1 and the schemes shown above. The whole beam path for either Bessel or Gaussian activation beam (marked as yellow in the pictures shown above) inside the cell can be imaged at the detection plane of the detection objective. As the reviewer pointed out, the Gaussian beam could form more confined photoactivation at the focus by excitation with a high numerical aperture (NA) objective. However, there is still substantial activation in the out-of-focus area, where optogenetic molecules in the cells could be activated in the beam path. Such a tightly focused Gaussian beam would create an uneven photoexcitation zone along the beam propagation direction inside the cell, whereas the Bessel beam provides an even activation zone within the cell. This was the reason why the Bessel beam was used for optogenetic activation in lightsheet microscopy instead of the Gaussian beam. Moreover, a Gaussian beam created by a high NA objective will result in limited lightsheet length, which cannot cover the whole cell for volumetric imaging. Such limited lightsheet length will become a serious issue as the sample size increases. For an imaging system with a similar lightsheet length, the Bessel beam has better confinement than the Gaussian beam. (For detail, please see Figure 1A in the cited reference 10, *Chen, B. C. et al. Science 346, 1257998, (2014)*). To help the reader to understand the properties of the Bessel beam and Gaussian beam excitation, we conducted a simulation on the lightsheet length, width, and intensity for both Gaussian and Bessel beams in the

supplementary notes and figure S9 and S10. To create an even photoexcitation for the Gaussian beam, a low NA objective should be used, where the length of the Gaussian lightsheet can cover the whole cell at the cost of the optical confinement, as shown in the middle of the above picture. Hence, the optical confinement/sectioning and lightsheet length become two crucial factors for optogenetic control using lightsheet microscopy. Moreover, to generate a lattice sheet in LLSM, an annular mask is used to filter out the unwanted diffraction from spatial light modulator (SLM) (please see the revised Figure 1 a), where the annular mask can be shared for Bessel beam generation but not for Gaussian beam. To clarify this point, we added the following paragraph in the supplementary note of the revised manuscript.

“Comparison of Bessel lightsheet and Gaussian lightsheet created by objectives with various NAs”

For the optogenetic experiment, the stimulation beam is used to activate the optogenetic molecules in cells. In principle, the stimulation beam can activate molecules in the beam path (except for two-photon stimulation). However, the detection schemes used in the experiment may create misleading images where all fluorescence signals along the propagation direction are projected onto a 2D image. To compare different excitation and detection schemes, the beam paths in the samples with different excitation schemes (wide-field, confocal, Gaussian lightsheet, and Bessel lightsheet) are depicted in Fig. S1. In figure S1a, a wide field excitation with a low NA objective is used where the fluorescence signals are detected by the same objective. With a high NA objective, a tightly focused beam is formed, as shown in figure S1b, where a pinhole in front of the detector can reject the out-of-focus signals forming confocal images. In both cases, the excitation beams activate molecules along the beam path, and the signals are projected onto a 2D image. For lightsheet microscopy, different objectives are used for excitation and detection. Shown in figure S1c and S1d are Gaussian and Bessel lightsheet illumination, respectively, where a separate detection objective at the right angle can image all the activated molecules along the beam propagation direction. To characterize the Bessel beam used in this experiment, we simulated and measured Bessel beam profiles in XY and XZ directions, as shown in figure S2. The point-spread function (PSF) along the beam propagation direction is calculated and measured at three spots marked in a white circle with a cross. In this experiment, maximum/outer and minimum/inner numerical apertures (NA) were 0.64 and 0.56, respectively. From the simulated and measured beam profiles, we can find that there are some intensity contributions from the side lobes. To compare with the light sheet created by the Gaussian beam, we need to quantify the intensity contribution from the main lobe and side lobes. We calculated the beam length, thickness, and effective intensity of the Bessel beam with various inner NAs at a fixed outer NA (=0.64). Shown in figure S9 is the beam profile of a Bessel beam near its focal point. The beam propagates along the y axis. We assume that the detection lens is located at the $+z$ side. Therefore, the field distribution on the xy plane is imaged. As indicated in Fig. S2, a Bessel beam is composed of the main lobe and many side lobes. The side of each lobe can be characterized by the full width at half maxima (FWHMs) along with the lateral and propagation directions, which are denoted as $FWHM_{x,n}$ (thickness) and $FWHM_{y,n}$ (length), respectively, where n represents n^{th} side lobe in the Bessel beam. Fig. S9b and S9c are the calculated $FWHM_{x,n}$ and $FWHM_{y,n}$ using the fast Fourier transformation under different sizes of ring apertures. The wavelength is $0.488 \mu\text{m}$. The outer numerical aperture NA_{out} defined as $\arctan(R_{\text{out}}/f)$ is set to 0.64, where R_{out} is the outer radius of the aperture, and f is the focal length. We calculate the thickness, length, and effective intensity of the main lobe and each side lobe at

a different inner numerical aperture $NA_{in} \equiv \arctan(R_{in}/f)$, where R_{in} is the inner radius of the aperture. For both the main and side lobes, the thicknesses ($FWHM_{x,n}$) are only reduced slightly as NA_{in} increases (thinner ring aperture) as shown in figure S9b. This is because $FWHM_{x,n}$ of the main and side lobes decreases with the average radius $(R_{out} + R_{in})/2$ of aperture, which is only varied slightly by R_{in} in our cases. The thickness of the main lobe is also larger than those of the side lobes, which is a feature of the zeroth-order Bessel function. On the other hand, the lengths of all the other side lobes ($FWHM_{y,n}$) increase significantly with NA_{in} (reduced aperture opening) due to the uncertainty principle as shown in Fig. S9c. Typically, the length of side lobes is prolonged more than the main lobe as NA_{in} increases, but their increment ratios do not differ much. To calculate the effective intensity I_n inside a rectangular region Ω_n defined by $FWHM_{x,n}$ and $FWHM_{y,n}$, we set the input power at $P_{in} = 1 \mu W$ at the ring aperture and calculate the effective intensity using the following equation:

$$I_n \equiv \frac{\frac{n_a}{2\eta_0} \int_{\Omega_n} d\rho |E(\rho)|^2}{FWHM_{x,n} \times FWHM_{y,n}}$$

where $n_a = 1.33$ is the refractive index of water (ambiance); $\eta_0 = 377 \Omega$ is the intrinsic impedance. As shown in Fig. S9d, the effective intensity I_n drops as NA_{in} increases as a result of the prolonged beam. In addition, the effective intensity of the main lobe is much higher than all other side lobes indicating that intensity contribution from side lobes of Bessel beams can be neglected in the Bessel lightsheet microscopy.

For comparison, we calculate the same beam properties for the Gaussian beam. As illustrated in Fig. S10a, the thickness, length, and effective energy of the Gaussian beam are calculated as a function of the numerical aperture $NA \equiv \arctan(W/f)$ of Gaussian beam, where W is the beam waist behind the excitation lens. As shown in Fig. S10b, unlike the thickness ($FWHM_x$) of the Bessel beam, the thickness of the Gaussian beam significantly decreases toward the diffraction limit as NA increases. On the other hand, the length of the Gaussian beam ($FWHM_y$) drops even more rapidly as NA increases (Fig. S10c). As the result of decreasing thickness and length as NA increases, the effective intensity shown in Fig. S10d exhibits tremendous enhancement as NA increases. The phototoxicity associated with such a high intensity may be problematic for living cell experiments”

2) Also, I’m unclear on the motivation for the maximum and minimum NA ’s of the Bessel beams chosen for photoactivation. Is there is a cutoff for minimum local intensity at the sample beyond which photoactivation will occur. How do the relative intensities of the main vs. side lobes of the Bessel beams factor into the lateral resolution estimates? Furthermore, I’m unclear how the resolution estimate of 20 microns along “y” was determined. Bessel beams (and Gaussian beams) can be generated with a much shorter beam length. Wouldn’t these provide better resolution along the propagation axis?

Response: The practical reason for choosing the Bessel beam instead of the Gaussian beam for photoactivation was the easy integration of the Bessel beam into the lattice lightsheet microscopy, where the SLM used for the generation of lattice lightsheet can be programmed to generate Bessel beam without additional optical components. In the lattice lightsheet microscope (LLSM) published in Chen et al. Science, 346, 1257998/1-1257998/12 (2014), reference 10, the annual mask was used for filtering out the unwanted laser diffraction. The layout of the mask is shown below, and the location of the mask is located

in the optical beam path after passing the SLM, as shown in the new Figure 1a. There are many annual rings with different outer and inner diameters matching different maximum and minimum NA's of the lattice and Bessel beams. In our design, the imaging lattice beam and the stimulating Bessel beam share the same annual pattern. Therefore, there is no need to change these annual patterns as the system is switched between imaging and photoactivation during the optogenetic experiment. However, the pattern used for the Gaussian beam is different (marked in red) from that of the lattice beam. If a Gaussian beam is used as the stimulating beam, an alternative optical path for the stimulation beam is needed. For easy integration of the photoactivation with LLSM, the same annual pattern, as shown below, was used for both LLSM and photoactivation. LLSM has been demonstrated capable of providing thin optical sectioning for the live 3D cell imaging by ultra-thin lightsheet generated from the high NA of the lattice beam, where a larger outer diameter of the annual mask was used. The thickness of the Bessel beam is governed by the maximum numerical aperture, whereas the length of the Bessel beam is determined by the minimum numerical aperture. In our system, we have chosen the largest NA in LLSM system, $NA=0.64$, which formed the Bessel main lobe with a thickness of $0.488/(2 \times 0.64) = 380$ nm, for the best optical sectioning. With this maximum NA, the minimum NA should be chosen such that the length of the lightsheet should cover the cell for 3D live imaging. However, there would be serious side lobe effects for such a high outer NA (0.64) due to the property of Bessel function. Therefore, to balance the length of the lightsheet and the side lobe, we conducted several experiments to optimize the maximum and minimum NA for photoactivation. The length of the lightsheet is crucial in the 3D imaging, where it should be long enough to cover the cell thickness for sample scanning mode in LLSM as shown in Figure 1b. To compare the effect of the NA on the length and thickness of the photoactivation, in the revised manuscript, we calculate the length, width, and the relative intensity of the main and side lobes of Bessel and the Gaussian beam in the supplementary notes, Figure S9 and S10. From the notes, it can be seen that the main lobe could be used for the sectioning where the energy density in the main lobe was much higher than those in the 1st or 2nd side lobes. The intensity and the width of the main lobe and side lobes used in our experiment can also be found in figure S3. In our configuration, the minimum energy required for photoactivation was about 0.2 nW.

20 micrometers along the “y” direction is not the imaging resolution. It is the length of the lightsheet as shown in the revised Figure 1b. From the scheme shown above, both Gaussian and Bessel beam activates molecules along the propagation direction, “Y.” The difference is that the Bessel beam provides a uniform activation in the “Y” direction, whereas nonuniform excitation is created by a Gaussian beam. In the lightsheet microscopy, the imaging plane is perpendicular to the excitation plane. The volumetric images consist of hundreds of 2D images (z-stack), where the separation distance in the z-direction is determined by the thickness of the lightsheet. To achieve high-quality volumetric images, the lightsheet illumination should be uniform within the cell, which indicates the whole thickness of the cell should be covered by the length of the lightsheet. Since the volumetric images were taken at a tilted angle about 32.8 degrees from the sample plane, as shown in Figure 1b, the length of the lightsheet should be larger than the thickness of the U2OS cell, which is about 10 micrometers. A lightsheet with 20 micrometers in length will be long enough to cover the whole cell for 3D imaging. If a Gaussian beam is used for volumetric imaging, an objective with $NA \sim 0.1$ needs to be used to create a lightsheet with the same length (20 micrometers), where the thickness of the lightsheet generated by such Gaussian beam is much thicker than the thickness of a Bessel lightsheet ($NA=0.64$) resulting in poor optical sectioning for 3D imaging.

The calculation of the thickness, length of the light sheet for both Gaussian and Bessel beams can be found in Figures S9 and S10 of the revised manuscript.

3) It would be very helpful if the authors could provide power estimates for imaging and photoactivation at the sample plane (e.g., W/cm^2) rather than just total power at the rear pupil of the excitation objective. Presumably the photoactivation efficiency is sensitive to the instantaneous local intensity rather than the total delivered dose.

Response: We thank the reviewer's suggestion, we reported the power measured at the sample plane in the revised manuscript. In line 27 and 122, "1 nW (or 24 mW/cm²)" added.

4) The authors do demonstrate some enrichment in photoactivation at either the apical or basal cellular membrane in Figure 2 by shifting the Bessel beam in "y" using a defocused pattern on the SLM, however, this isn't really "precise 3D activation" as it's achieved by shifting a rather long activation beam beyond the physical extent of the sample rather than by confining it to a specific cellular region. Moreover, no quantification of the relative activation at the apical or basal membranes is provided and Figure 2 appears to show only a very minor difference between the "Bessel fan" and "shifted Bessel fan" conditions. Quantifying the axial activated area (similar to as was done for figure S2) would be helpful.

Response: We thank the reviewer for the helpful suggestion. By shifting the beam position, we could activate the optogenetic molecules on the apical or basal cellular membrane. The difference between the Bessel beam and the shifted Bessel beam activation is their axial position along with the axis of the excitation objective. For more precise photoactivation in a specific cellular region, the single Bessel beam can be used where only a local volume can be activated, as shown in the new figure S12. As figure S1 illustrates that all types of stimulations except the two-photon photoactivation will activate molecules in the beam path inside the cell. Changing the beam distribution will only increase the signal-to-noise ratio of the activated molecules at a local volume where the diffusion will reduce such local activation.

Therefore, we agree with the reviewer's concern about the use of "precise 3D activation", which was removed in the revised manuscript. To see the difference between different photoactivation processes, we added quantitative analysis of the fluorescence signals in the revised manuscript, where the time-dependent intensity changes of the CRY2oligo-mRuby3 aggregations in three different cases (Bessel fan, shifted Bessel fan, and single Bessel beam) were included as figure 3 and supplementary figure S5-6. For the single Bessel beam activation case, the results are shown in figure 3. A rapid increase in the intensity of CRY2oligo-mRuby3 aggregate at the photoactivation region can be clearly seen (cluster #1). A delayed intensity increase (10 s) was observed for a cluster near the photoactivation region (cluster #2), which was 2 micrometers away from the activation region, indicating initial photoactivation was localized both in space and time. However, due to diffusion, the photoactivation propagated in the cytosol. The intensity of a cluster (cluster #3) 5 micrometers away from the activation region increased around 40 s after photoactivation. To demonstrate that the photobleaching effect during the imaging period was not critical, we chose a pre-existing fluorescent cluster (cluster #4) far away from the activation region and observed the change in fluorescence intensity over time of this cluster. No significant change in intensity was recorded. In the case of "Bessel fan" activation shown in Figure S5, the fluorescence intensity of the clusters (#1-#2) near the "Bessel fan" activation zone increased over time, whereas the clusters far away from the activation zone, such as # 3-5 exhibited a delayed and weak response according to their distance from the activation zone. A strip of fluorescent clusters was observed across the cell as the result of the "Bessel fan" activation. In the shift Bessel beam case shown in Figure S6, clusters close to the basal layer exhibited an immediate response for the activation near the basal layer, such as #1, whereas delayed responses were observed for clusters away from the basal membrane where the delay time depended on the distance from the basal membrane. The movies for all three activation cases can be found in the supporting materials as movies 3,4,5,11, and 12.

Minor

comments:

1) Pg 7 "Each imaging volume consisted of 131 stacks of images, which took 2.6s to collect". I'm assuming this is meant to say "131 2D images" rather than "131 stacks of images".

Response:

We agree with the reviewer's suggestion. In the revised manuscript, the sentence has been rewritten as "Each imaging volume consisted of 131 planes." In line 115.

2) "RFP" is used as a general abbreviation for "red fluorescent protein" which is then followed by a specific description of the fluorophore-conjugate (e.g. F-tractin-mCherry). This is confusing as RFP is generally an abbreviation for the specific red fluorescent protein (e.g. mRFP). It would be better just to list the specific proteins used.

Response:

We agree with the reviewer's suggestion. In the revised manuscript, the RFP has been replaced by the corresponding fluorophore conjugate.

Reviewer #2 (Remarks to the Author):

Major claims of the paper:

In this paper, the author claim to produce controlled and local (in 3D) activation of optogenetic tools. They shows that local recrutement to the plasma membrane of a iSH domain(activator of Cdc42/Rac1) promotes membrane ruffling and the the local formation of EGFR clusters promotes cell migration.

Novelty of the claims:

To my reading, this paper is meant to promote the use of LLSM to do local and long term optogenetic experiment. Then, the two biological experiments seem to be present just as a validation of the use of this microscopy setup.

I have a few comments about this novelty:

First of all, and not mentioned/referenced in the main text, researchers are now doing optogenetic experiment, with subcellular (~2-4um precision) control of activation, for more than 10 years, without the need of LLSM. They indeed used a wide range of setup from spinning disc+ frap modules, widefield microscope +dmd illumination setups, to scanning laser confocal microscopes+ frap routines, and most of the time managed to do experiment ranging over 12h experiments. For even better resolution, people already used some 2 photon microscopes coupled to optogenetic experiments and/or used optogenetic systems enabling the control of inactivation. This has to be mentioned.

Response: We thank the reviewer's suggestion. In the revised manuscript, we included the achievement of optogenetic experiments by conventional microscopic tools. More references were added, including the optical/optogenetic experiments with subcellular control Taslimi et al. Nat Commun. 2014 Sep 18;5:4925 (ref 25); Bugaj et al. Nat Commun. 2015 Apr 22;6:6898 (ref 31); Karunarathne et al. PNAS. 2013 Apr 23;110(17):E1575-83 (ref 1); Kim et al. PNAS, 2016 May 24;113(21):5952-7 (ref 8), the application of spinning disc for the optogenetic experiment, John Allen Nature Methods 14, page1114 (2017) (ref 6), Johnson et al. Dev Cell. 2017 Jan 23;40(2):185-192 (ref 7), widefield with DMD illumination, tow-photon activation in the optogenetic experiment, Xu et al. Proc Natl Acad Sci U S A. 2014 Apr 29;111(17):6371-6. (ref2), Ronzitti et al, Journal of Optics 19, (2017) (ref 5). The following sentences were added in the introduction of the revised manuscript to mention the utilization of other optical tools for the optogenetic experiments and the justifications for using LLSM for the optogenetic experiment. "Spatiotemporal manipulation and recording of biological processes, such as cell migration or membrane dynamics with the subcellular resolution, is crucial in understanding these fundamental cellular processes^{1,2,3}. However, it is technically challenging for cell manipulation with a precise spatiotemporal resolution, especially in three dimensions (3D)^{4, 5}. A common approach for subcellular manipulation can be achieved by optogenetic tools on a conventional microscope such as wide-field or confocal microscope^{6,7,8}. Recording the three-dimensional rapid cellular response at high spatiotemporal resolution can be problematic, where a high numerical aperture (NA) objective is often used to create a tightly focused spot for better confinement of the photoactivation beam. When such a photoactivation beam propagates in the cell, it

activates molecules not only in the focal spot but also along the beam path where the out-of-focus activation may not be imaged when a pinhole is used to reject the out of focus signal as in the case of confocal microscopy (the beam paths of different excitation and detection schemes can be found in Figure S1)⁹. As a result of tight focusing, an uneven photo-excitation along the beam propagation is created. Therefore, it is desirable to develop a wide-field technique with good sectioning capability to record the optogenetic response. A notable implementation in this regard is the development of lattice light-sheet microscopy (LLSM), which is constructed by an array of Bessel beams, where destructive interference of the coherent Bessel beams minimizes the contribution from sidelobes within the detection zone¹⁰. LLSM provides several advantages for high-resolution volumetric imaging of living cells, including reduced phototoxicity in which the excitation energy is distributed over an array of the illumination beams, an optimal balance between the thickness and length of the lightsheet, enabling high spatial resolution imaging with a good optical sectioning capability, and improved signal-to-noise ratio in which the out-of-focus noise is suppressed by the coherent modulation of lattice beams^{11, 12, 13}. In addition, the lattice patterns also allow easy implementation of super-resolution microscopy using structured illumination¹⁴.”

Moreover, in order to emphasize the long-term imaging with a high temporal resolution for LLSM imaging, we added the following imaging acquisition information for the presented 6 h optogenetic experiments, in line 29, “using optogenetic stimulation for up to 6 h, where 463 imaging volumes are collected,” In line 253, “more than 6 hours, where ~75,000 2D-images (or 463 volumetric images) were collected,”

Now concerning this article

First, the LLSM setup and the possibility to do local illumination have already been described in reference 23.

Response: Yes! The local illumination has been described in reference 23, proceedings of microscopy and microanalysis, now ref 26 in the revised manuscript. However, the concept of photoactivation was not clearly explained. As mentioned in the introduction, the invention of lattice lightsheet microscopy (LLSM) since 2014 by Chen, B. C. et al. *Science* 346, 1257998 (Ref.10) has drawn a lot of attention for live cell imaging with high spatiotemporal resolution. Since the LLSM setup is sophisticated, our attempt in this experiment is to report an easy integration of the photoactivation through spatial light modulator (SLM) without changing the optical setup. As demonstrated by our group as the first proof-of-principle experiment in ref 23 (now ref 26), a carboxylate-modified fluorescent microsphere was used to prove the capability of moving “two” directions, x and z, by SLM patterns. With the idea of defocusing the profile on the SLM patterns, we were able to control the lightsheet beam in the x,y,z direction in the optogenetic experiment. To clarify this point, Figure 1 was revised to better illustrate the light sheet illumination, and the description in the result section was rewritten as on page 6 line 92, “To conduct an optogenetic experiment in LLSM , our system consists of a large parallel array of coherently interfering Bessel beams through the SLM in which individual Bessel beams with selected wavelengths can be used as activation sources by simply changing the patterns on the SLM²⁶ located at the plane conjugated to the imaging plane (Figure 1a and 1b). With a different pattern on the SLM, the lattice beams at the imaging plane can

be switched to a stimulating Bessel beam whose x, y positions can be controlled by different modulations in the SLM patterns, as shown in Figure 1b.

Second, the author claims that they do local optogenetic activation, however, due to the diffusion properties of the optogenetic tool, the “local” appears on most of their movies “almost global”. This could be improved by better quantification (more precise, and with statistics on more cells).

Response: We thank the reviewer’s suggestion. We agree that the diffusion of the activated molecules may lead to global activation. To quantify the photoactivation processes, we calculated the fluorescence intensity of the photoactivated clusters in different parts of cells for three types of photoactivation processes. The results are shown in figure 3 and figure S5 and S6 of the revised manuscript. As mentioned in the response for reviewer 1, a rapid increase in the intensity of CRY2oligo-mRuby3 aggregate at the photoactivation region can be clearly seen (cluster #1). A delayed intensity increase (10 s) was observed for a cluster near the photoactivation region (cluster #2), which was 2 micrometers away from the activation region, indicating initial photoactivation was localized both in space and time. However, due to diffusion, the photoactivation propagated in the cytosol. The intensity of a cluster (cluster #3) 5 micrometers away from the activation region increased around 40 s after photoactivation. To demonstrate that the photobleaching effect during the imaging period was not critical, we chose a pre-existing fluorescent cluster (cluster #4) far away from the activation region and observed the change in fluorescence intensity over time of this cluster. No significant change in intensity was recorded. In the case of “Bessel fan” activation, the fluorescence intensity of the clusters (#1-#2) near the “Bessel fan” activation zone increased over time, whereas the clusters far away from the activation zone, such as #3-5 exhibited a delayed and weak response according to their distance from the activation zone. A strip of fluorescent clusters was observed across the cell as the result of the “Bessel fan” activation. In the shifted Bessel beam case, clusters close to the basal layer exhibited an immediate response for the activation near the basal layer, such as #1, whereas delayed responses were observed for clusters away from the basal membrane where the delay time depended on the distance from the basal membrane. The movies for all three activation cases can be found in the supporting materials as movies 3,4,5, 11, and 12. Following sentences were added in the revised manuscript to describe the characterization of the spatiotemporal behavior of the photoactivated clusters on page “To analyze the spatiotemporal behavior of the activated clusters, 3D cell volume renderings with selected CRY2oligo-mRuby3 clusters are shown in Figure 3a, where cluster 1 is located right at the activation region, clusters 2 and 3 are 2 μ m and 5 μ m away from the activation region, respectively. The time-dependent fluorescence intensity changes of the chosen clusters are plotted in Figure 3b, where cluster 1 shows an instantaneous increase upon the activation beam. The delayed responses are observed according to the distance from the activation spot, for example, 10 s for cluster 2 and 40 s for cluster 3. To evaluate the photobleaching effect in our system, we monitored the fluorescence signal of a pre-existing cluster (cluster 4), which was far away from the activation beam. No decrease in fluorescence intensity of cluster 4 was observed during the whole experimental period indicating that the photobleaching effect of our system is negligible. A series of raw images for the selected plane activated by a single Bessel beam are shown in Figure 3c, where the formation of the cluster over time is marked in a white box. The analysis of fluorescence intensities of different clusters activated by Bessel fan and shifted Bessel beam is shown in Figures S5 and S6. These findings clearly demonstrate that LLSM could be easily modified to conduct 3D optogenetic experiment with good spatial control using a regular and modified Bessel fan/beam as an activation source.”

To demonstrate that our approach can locally activate the response of the cells, we conducted a membrane ruffling experiment (figure 4) and directed cell migration experiment (figure5). From our results, we clearly showed that the activation could be localized in a small part of a cell.

Third, the 2 validation experiments are interesting. However, the “local” activation ends up to be very broad. Then, it might be better to do a deeper characterization of the processes involved here which are so far very descriptive.

Response: We thank the reviewer’s suggestions. As for the characterization of the behavior of the activated molecules, we added the time-dependent fluorescence intensity of the activated molecules in different parts of the cell, as in figure 3 and S5-S6 in the revised manuscript. We also rendered and quantified the volume imaging of the guided cell migration as new figure 5 and movie 10. In line 247, “The individual LLSM rendering images of the 3D cell and the stimulated area are depicted at the bottom of Figure 5 and Movie 10. Moreover, the volumes of the migrating cell over time are calculated and plotted in the inset, where the volume of cell reduces when migration direction changes at T=265, followed by the formation of large lamellipodia at T=292.” In the inset of figure 4a in the revised manuscript, we also show the speed of the membrane process during activation. In line 207, “white cross in figure 4a region 1), the membrane ruffled around the area under stimulation and moved inward at the speed of ~1 $\mu\text{m}/\text{sec}$ where the moving distance of the membrane edge along the white arrow is plotted in the inset.” In Figure S8, we used polar coordinate to quantify the amount of protrusion and retraction at different parts of the cell with respect to the geometric center of the cell. In line 218, “The membrane protrusion and retraction upon the activation, as shown in Figure S8, reveals that the membrane ruffling is presumably increased at the site of activation as compared to the rest of the cell body.”

Overall, I find that the title is an overstatement on many points and that, presented this way, this article is just a small technical adaptation of already existing/published tools. Validity of the results: Concerning the validity of the result, I do trust the implementation of a Bessel beam. However, concerning biological experiments, we do not have access to how reproducible are the experiments.

Response: We thank the reviewer’s suggestion, we change the title of our manuscript to “Optogenetic Manipulation of Cell Migration with High Spatiotemporal Resolution Using Lattice Lightsheet Microscopy.” We report here an important feature for the LLSM for conducting optogenetic experiments. We believe that the high-resolution fast volumetric imaging will be very helpful for researchers in conducting optogenetic experiments. Indeed, Zeiss company introduced their Lattice Lightsheet microscope (<https://www.zeiss.com/microscopy/int/products/imaging-systems/lattice-lightsheet-7.html>). Moreover, there are many clones as a imaging core scattered over the world such as AIC at Janelia, Harvard, UCSF, RIKEN at Japan, Vienna BioCenter Core, Peking University, and so on. As for the reproducibility of our experiment. This will definitely broaden the usage of LLSM and shed new light on biological research.

Moreover, there is no clear quantification of how spread is the activation, what are the different time courses, ...

Response: We thank the reviewer’s suggestion, In the revised manuscript, we added the time-dependent fluorescence intensity at different parts of cells as Figure 3 and S5-S6, which can be used to evaluate the

spread of the activation. The movies of these activated cells can be found in the supplementary information as described in the previous answers.

Writing:

Last but not least, many parts of this article, starting with the abstract, should be rewritten to improve clarity.

Finally, figures are often missing a lot of information of interest (as for the scheme fig1a) and there captions should be improved.

Response: We thank the reviewer's suggestion, we rewrote our manuscript to improve the clarity, and we added more description in the figure captions to help readers understand the concept of our design.

All this together, I do not recommend this article for publication

Response: We hope that we addressed all issues raised by the reviewer, and we believe that our revised manuscript is suitable for publication in Communications Biology. Last but not least, we were not meant to say our technique (LLSM) is better than the other (confocal). We tried to offer an alternative option/view to look at the biological process to help us to advance our understanding of the complicated biological processes. Hopefully, all the answers could remove or reduce the concerns from the reviewer.

Reviewer #3 (Remarks to the Author):

Summary: Tang et al. describes a method of performing sub-cellular photoactivation in conjunction with lattice light-sheet microscopy (LLSM). Most notably, the authors perform these experiments without use of any additional modules or alterations to the original LLSM design through alterations of the spatial light modulator (SLM) patterns. To test their method, the authors developed two probes: (1) CRY2olig (optically induced clustering) and (2) CRY2-CIBN (triggering of regional membrane ruffling). Optically induced clustering experiments were used to demonstrate “Bessel Fan” photoactivation both throughout the entirety of the cell body and at the cell-substrate interface, as well as single Bessel beam photoactivation. Triggering of regional membrane ruffling was used to showcase the method’s ability to dynamically change the site of activation in shortterm and long-term experiments. The authors clearly demonstrate that they are able to perform sub-cellular photoactivation using only the original LLSM design, which could be of great interest to the microscopy and biology communities. However, I have further questions regarding the implementation methodology, photoactivation precision, and image quantification.

Major Comments:

1. Given that movement of the beam position through alterations in the SLM pattern (doi:10.1364/OE.27.001497, cited in this work) and photoablation/FRAP/photoactivation with LLSM (https://doi.org/10.1083/jcb.201906111, https://doi.org/10.1101/2020.09.01.276824,

<https://doi.org/10.1117/12.2509467>) have been previously shown, the novelty of this work is in the use of only the original LLSM design to perform sub-cellular photoactivation. This is important because it dramatically improves the ease of implementation and accessibility of the method for other groups with access to LLSM. However, this makes accurate reporting of the method of highest importance, and I feel as though the manuscript is currently lacking in sufficient detail.

Response: We appreciate the reviewer's comment to highlight the importance of our technique for the LLSM community in which no additional complicated optical components are needed in the optical path. The photoactivation can be achieved by simply altering the binary patterns onto the SLM. In the revised manuscript, we rewrote our manuscript to highlight the strength of our technique and included the references as suggested by the reviewer. In line 60, "With an additional optical path for illumination, it has been demonstrated that photoablation/FRAP/photoactivation experiments^{18, 19, 20} can be conducted on LLSM. However, it remains challenging to perform experiments requiring optical stimulation, such as subcellular optogenetic applications, in LLSM."

Specifically:

a. For single Bessel beam activation, how is illumination for a single slice practically achieved? It appears as though the activation is performed during a z-stack, so how is the light-gated for the rest of the acquisition? The authors make reference to a mechanical shutter, but it is not evident how this is controlled dynamically during an experiment.

Response: For single Bessel beam activation, the activation beam was illuminated on the samples at a selected z slice while performing z-stacking. A mechanical shutter placed in front of 488nm laser was used to allow the activation beam to pass through the sample at a specific time, which was controlled by a homemade Arduino device synchronized to the image acquisition software. The detailed protocols are included in the method of the revised manuscript, line 394 "with a mechanical shutter which is placed in front of the 488 nm laser, controlled by a homemade Arduino device synchronized to the image acquisition software. For single Bessel beam stimulation, the SPIM program held the x-galvanometer without any movement, thus allowing the Bessel beam to stimulate the central area of the cells, and the activation wavelength 488 nm was illuminated on the samples at a selected z slice while performing z-stacking; the timing is clocked by Arduino device."

In addition, to illustrate the activation and imaging process through the 3D sample scanning, the illumination sequences for different activation schemes, where the SLM patterns and laser colors are changed accordingly, are now included in the revised figure 2 and line 388 "As shown in the bottom of figure 2a, the SLM displaying two pre-loaded color patterns (488 nm and 560 nm) for either Bessel/lattice beam or shifted Bessel/lattice beam at each step of the sample stage at the designed time sequence based on the experimental exposure time in the SPIM software."

b. For experiments where the activation beam position is changed during the experiment, how is this performed? It is evident that the position change is done by changing the SLM pattern, but is this done dynamically or does the experiment need to be paused to upload a new pattern to the SLM?

Response: The main concept of our design is to conduct photoactivation through the SLM, where the corresponding patterns for different photoactivation schemes can be preloaded to the SLM. During the experiment, the illumination sequence and SLM patterns are shown in Figure 2. This type of experiment can be done automatically. However, for cell manipulation, such as induced membrane ruffling and directed cell migration experiments, the subcellular responses of cells cannot be predicted. Therefore, we cannot program the SLM pattern time profile in advance. To demonstrate the capability of local activation, the photoactivation was conducted manually. After recording the first cellular response, the program was stopped, and the activation beam was moved to the desired area either by changing the SLM pattern or the x-galvo position. The program was resumed to record the cellular response. These processes were repeated until the end of the experiment. In line 420, we added the following information, “Note that for cell manipulation, such as induced membrane ruffling and directed cell migration experiments, the subcellular responses of cells cannot be predicted. Therefore, we cannot program the SLM pattern time profile in advance. To demonstrate the capability of local activation, the photoactivation was conducted manually. After recording the first cellular response, the program was stopped, and the activation beam was moved to the desired area either by changing the SLM pattern or the x-galvo position. The program was resumed to record the cellular response. These processes were repeated until the end of the experiment.”

c. In the directed migration experiment, the authors reference the need to reposition the sample every 20-30 minutes. Does this imply that the acquisition was paused to move the sample and then restarted? How is the precise acquisition timing and sample position monitored during the experiment?

Response: Yes! We did stop the experiment for 10 seconds to reposition the sample. In our setup, the field-of-view of LLSM was about $\sim 100 \times 100 \times 20$ μm at the pixel resolution of about $\sim 100\text{nm}$. For the long-term and directed cell migration experiments, the cell would probably move out of field-of-view. Without the automatic XYZ stages in our setup, we had to manually move the sample back to the imaging center of the field of view. Since the time course for this long-term experiment was about 6 hrs, the repositioning time is negligible, which wouldn't affect the observation for the directed cell migration. However, the timing and position were registered for future reference in the movie reconstruction. In line 243, “adjusted while maintaining the same stimulated area in the cell, as shown in Movie 9.” In line 430, “For the directed cell migration, the timing and position were registered in the movie reconstruction, as shown in figure 5 and movie 10.”

d. Overall, the heart of this paper seems to lie in the description of how the authors achieve sub-cellular photoactivation in the conventional LLSM system, for which the description is minimal. I would recommend for each specific variant (Bessel Fan, Y-Shifted Bessel Fan, Static Single Bessel, Dynamic Single Bessel), the authors should allocate a describe in detail how each of these experiments were practically realized. This is necessary to ensure reproducibility, and at present these experiments would not be reproducible by a group with access to LLSM.

Response: We thank the reviewer for the helpful suggestions. In the revised manuscript, the illumination scheme over z-stack scanning was added to figure 2. In addition, highlighted in the revised manuscript, we have added more experimental details in the material and method part, and the detailed optical scheme is shown in the supplementary note and figure S11 for reference; in line 708 “The schematic of

the optical system is shown in Fig. S11. The beam from a laser combiner equipped with 488 nm (300mW, Coherent Sapphire 488 nm 300-CW), 561 nm (200mW, Oxxius LMX-561S-200-COL-PP) lasers is expanded to a diameter of 4 mm by two lenses (8 mm FL/ Ø1/2", Thorlabs C240TME-A, 20 mm FL/ Ø1/2" Edmund 47-661). The exposure time and the wavelength selection can be controlled by an acousto-optic tunable filter (AA Quanta Tech, Optoelectronic AOTF AOTFnC-400.650-TN) (1).

A pair of cylindrical lenses (Edmund NT68-160, 25 mm FL/12.5 mm dia (2); Thorlabs, ACY254-250-A (3)) is used to expand the beam in x axial direction. The expanded beam then passes through a polarizing beam splitter cube (PBS, Newport, 10FC16PB.3) (4) and a half-wave plate (Bolder Vision Optik, BVO AHWP3) (5), and uniformly illuminates on the central region of the spatial light modulator (SLM). The SLM itself consists of 2048 × 1536 ferroelectric liquid crystal pixels (Forth Dimension, QXGA-3DM) (6), which can change the polarity of the diffracted beam depending on the state of each pixel. The polarized beam can be imaged onto a custom quartz mask (8) by a polarizing beam splitter (4) cube and a lens (Edmund, 350mm FL / 50mm dia, VIS-NIR coating, achromatic lens (7)).

The lens pair (Thorlabs, AC254-100-A (9) and AC254-075-A (10) Ø1" Achromat, 400 - 750 nm) can reduce and image the beam from the mask to combine with Z axial galvanometer scanner (11). A relay lens (Thorlabs, AC254-85-A (12 and 13) Ø1" Achromat, 400 - 750 nm) combines two galvanometer scanners in the Z-axis (11) and the X-axis (14). After passing through two-dimensional scanning mirror sets, the beam is magnified through a relay lens (Thorlabs, AC254-254-A (15) and AC254-400-A (16) Ø1" Achromat, 400 - 750 nm) and conjugated to the back focal plane of the excitation objective (Special Optics, 0.66 NA, 3.74 mm WD) (17). The beam is projected onto the back focal plane of the excitation objective, and a self-reconstructed lattice beam is formed by optical interference at an incident angle of 32.8 degrees to the coverslip. Orthogonal to the illumination plane, water immersed objective lens (Nikon, CFI Apo LWD 25XW, 1.1 NA, 2 mm WD) (18) mounted on a piezo scanner (Physik Instrumente, P-726 PIFOC) (19) is used to collect the fluorescence signal, which is then imaged through an emission filter (Semrock Filter: FF01-523/610-25 and FF01-446/523/600/677) onto an sCMOS camera (Hamamatsu, Orca Flash 4.0 v2 sCOMS) (21) by a 500 mm tube lens (Edmund 49-290, 500 mm FL/50 mm dia; Tube Lens/TL) (20)."

In Figure 2a, the timing diagrams for the SLM patterns with laser 488 nm Bessel beam (blue) and 560 nm lattice beam (green) illumination were included as well as the sample stage synchronization waveforms (black). In line 141, we added "The corresponding timing diagrams for switching on the activation (blue) and imaging (green) laser by two-color SLM and the movement of the sample stage (black) are illustrated at the bottom of figure 2a."

In the materials and methods of the revised manuscript, in line 414, the following sentences were added, " The camera settings for the directed cell migration experiment are the same (10 ms exposure time for 200 x 1024 pixels with 161 planes) for both colors 488 nm (activation) and 560 nm (imaging), where it takes ~ 3 s to finish 3D optogenetic imaging. Initially, the time interval between each volumetric image was set to 6 s until we observed cellular response or membrane ruffling (in this case, 60-time points). We then changed the time interval to 1 min to observe directed cell migration for 343-time points. Note that for cell manipulation, such as induced membrane ruffling and directed cell migration experiments, the subcellular responses of cells cannot be predicted. Therefore, we cannot program the SLM pattern time profile in advance. To demonstrate the capability of local activation, the photoactivation was conducted manually. After recording the first cellular response, the program was stopped, and the activation beam was moved to the desired area either by changing the SLM pattern or the x-galvo position. The program

was resumed to record the cellular response. These processes were repeated until the end of the experiment.”

2. One major claim repeated throughout this paper is that the necessary photoactivation power is on the order of a single nW. Can the authors provide a more in-depth description of how these measurements were performed? The original LLSM paper used excitation powers on the order single μ Ws at the lowest, and this work claims to perform volumetric imaging at 40 nWs. A 100 fold change in power seems highly unusual, and the power range of the power meter referenced ends at 0.5 nW, but the authors reference using powers of 0.2 nW which in principle cannot be determined with the referenced instrument. Can the authors please clarify exactly how these measurements were carried out and validated?

Response: We thank that the reviewer raised this issue. For the activation power, the biological system (CRY2oligo-mRuby3) was very sensitive to the light activation, which is on the order of \sim nW based on our experimental results. In some cases, we found that even the light from the monitor (\sim 0.17nW according to the spec of the monitor) could trigger the photoactivation. Therefore, very low 488nm activation energy was used in our experiments. As pointed out by the reviewer, the detection limit of the power meter is about \sim 0.5 nW. In our experiment, we first measured the power above the detection limit of the power meter, and then the laser power was attenuated linearly through the acousto-optic tunable filter (AOTF) and neutral density filters in front of 488nm laser. In such a way, we determined that the minimum activation energy was 0.2 nW. In the revised manuscript, the more detailed experimental protocol for measuring the activation energy was included in line 382. “To measure the laser power below the detection limit of the power meter, which is about 1 nW, the laser power was attenuated linearly by a combination of the acousto-optic tunable filter (AOTF) and the neutral density filters.”

Regarding the power we used for volumetric imaging, which was about 40 nW in our experiment, several factors need to be considered for choosing the right power, including the brightness of the fluorophores, the needed signal-to-noise, the overall data acquisition time and etc. Since we are working on photoactivation experiments, we tried to avoid the use of high power for volumetric imaging because of the possibility of activation by cross-talk, where the activation could be induced by 561 nm at high power. To conduct lower power activation, we have constructed a brighter red fluorescent protein 3x mRuby protein and used a signal-to-noise ratio of about two, where the cell morphology could be barely distinguished. That’s why a very low power (40 nW) was used for 3D cell imaging compared to a typical value (μ W) reported in the published LLSM papers. In the revised manuscript, we have highlighted this difference as follows in line 230 “By maximizing the brightness of red fluorescence protein for whole-cell imaging, the cross-talk activation could be minimized due to a very low dose laser power.”

3. Any microscopy development is subject to a tradeoffs, and it is important to discuss these tradeoffs to help a reader weigh the pros and cons of implementing a technique. The authors have done well to highlight many of the strengths of their methods, and have begun discussing limitations at the end of the discussion section, but I think it would be beneficial to expand this discussion. For example, the annular mask cannot be dynamically switched during an experiment in the conventional LLSM system. Therefore, the inner and outer NA of the Bessel beam used for photoactivation is predetermined and the length of the beam will be similar to that of the lattice used for imaging. This means that the localization of photoactivation in the direction of propagation is somewhat limited by the depth of field needed for

image acquisition. An expanded discussion of this limitation and others, contextualized with the benefits of the method, is necessary.

Response: We thank the reviewer's suggestion. In the revised manuscript, we have added the following statements in line 319 to discuss the limitations of LLSM using SLM for photoactivation. "To improve the spatial confinement for activation, two-photon activation or temporal focusing may be used^{35,36}. However, the integration of these techniques into LLSM may be complicated. Another challenge in implementing 3D activation in LLSM was the line confinement activation instead of a point in the propagation direction, which was about 20 μm at the present configuration in order to cover the whole cell imaging. Such confinement allows us to conduct local activation of molecules located on the apical or basal membrane by SLM manipulation but not enough to activate molecules within a specific organelle in the cells with the fixed annual mask. An aperture-free technique for the generation of activation beam and lattice beam at different beam lengths is required in the future. In summary, the 3D activation capability of our approach opens the opportunity for optogenetic activation of single cells in small animal models."

4. It is difficult to assess the specificity of the photoactivation given the diffusion of CRY2oligo-mRuby3 (as noted by the authors) and the nature of membrane ruffling itself. Because this paper fundamentally relies on sub-cellular photoactivation, I think it is important to have an accurate description and quantification of the area being activated. To assess this, I would suggest the authors perform a photobleaching assay on a fixed sample using each variant of photoactivation approach (Bessel Fan, Y-shifted Bessel Fan, Static Single Bessel, Dynamic Single Bessel) described in this paper (though the light intensity may need to be modulated to achieve significant photobleaching). This would allow the authors to show precision and accuracy of the photoactivation approach described in this manuscript.

Response: We thank the reviewer's suggestion for the photobleaching experiments. Since we are focusing on live imaging, we decided to show the spatiotemporal control in the live samples as shown in the new Figure 3, S5, and S6, where we calculated the time-dependent fluorescence intensity of several activated clusters in different parts of cells for three different activation schemes (Bessel fan, shifted Bessel beam and single Bessel beam). From all these results, a minimum photobleaching effect was observed in our experiment. In the revised manuscript, the following sentences were added to illustrate these results in line 172, "To evaluate the photobleaching effect in our system, we monitored the fluorescence signal of a pre-existing cluster (cluster 4), which was far away from the activation beam. No decrease in fluorescence intensity of cluster 4 was observed during the whole experimental period indicating that the photobleaching effect of our system is negligible."

5. Many of the conclusions of this study appear to be derived from representative examples as opposed to through quantification over number experiments. Specifically, the following conclusions are drawn without quantification of the associate images:

a. The minimum power for photoactivation is 0.2 nW.

Response: As mentioned above, we measured the power of the photoactivation in the back pupil plane and the sample plan. We have added the associated information in the method part of the revised

manuscript in line 383 “ To measure the laser power below the detection limit of the power meter, which is about 1 nW, the laser power was attenuated linearly by a combination of the acousto-optic tunable filter (AOTF) and the neutral density filters.”

b. Shifted Bessel Fan excitation concentrates photoactivation at the cell-substrate interface.

Response: In the revised manuscript, we added the quantitative analysis of the intensity of activated clusters at a different location along the direction of cell-substrate interfaces. New results were included in the revised manuscript as shown in Figure S6, in line 780, “Characterization of the spatiotemporal behavior of the photoactivated clusters of CRY2oligo-mRuby3 expressed in the cell (a) illuminated by shifted Bessel fan photoactivation schemes. (b) the time-dependent fluorescence intensities for the clusters marked in (a). (c) the z slice images of the locations for the marked clusters among the 131 slices with z interval of 0.6 μm .”

c. membrane ruffling is increased at the site of activation as compared to the rest of the cell body

Response: We thank the reviewer pointed this out. We have added the quantification of the membrane dynamics such as moving distances over time as shown in the inset of the revised Figure 4a region 1, in line 206, “After stimulation (white cross in figure 4a region 1), the membrane ruffled around the area under stimulation and moved inward at the speed of $\sim 1 \mu\text{m}/\text{sec}$ where the moving distance of the membrane edge along the white arrow is plotted in the inset.” We also add Figure S8 to show the membrane dynamics for the movement inward and outward, which shows the big change for the activation area compared to the rest of the cell body, in line 800 “Quantification of the membrane protrusion and retraction during the guided cell migration. A polar coordinate is used to quantify the amount of protrusion and retraction at different parts of the cell with respect to the geometric center of the cell before and after photoactivation.”

d. Directed migration can be induced through localized photoactivation.

Response: In revised figure 5, the 3D images of the cell during the cell migration as well as the locations of photoactivation relative to the cell body are included in the revised manuscript. We also calculated the changes in cell volume at different time points. Through these data, we reached the conclusion that guided migration can be induced through local photoactivation. “The individual LLSM rendering images of the 3D cell and the stimulated area are depicted at the bottom of Figure 5 and Movie 10. Moreover, the volumes of the migrating cell over time are calculated and plotted in the inset, where the volume of cell reduces when migration direction changes at $T=265$, followed by the formation of large lamellipodia at $T=292$.” In the new movie 9, showing at the beginning without activation, the cell remains steady; after activation, cell membranes ruffle dynamically and eventually move toward the guided direction shown in line 933 and movie 9, “The Raw 2D images used to reconstruct 3D volumetric image in the Bessel fan photoactivation experiment, where CRY2oligo-mRuby3 molecules are stimulated by 488 nm across the whole cell. 488 nm can excite the activated CRY2oligo-mRuby3 clusters with low efficiency, where the weak fluorescence signals reveal the contributions of main and side lobes of the illuminated Bessel beam

forming a stripe of a width of 2 \$\mu\$ m (left) as opposed to a lattice sheet imaging by 560 nm (right). Note that the residual electronic signal in the sCMOS leaking from the 560 nm channel was observed at the 488 nm channel."

The data presented by the authors is consistent with the conclusions that they have drawn, however, it is not sufficient to state conclusions from single example. Further repeated experiments and quantification are necessary to show these statements as fact. Otherwise, the authors must emphasize that these are simply observations from single examples as opposed to conclusions from repeated studies.

Response: We thank the reviewer's suggestion, in the revised manuscript, we added more experimental data and quantitative analysis to support our claims as new figures 3, S5, S6, S8, S12.

Minor Comments:

1. The authors should make reference to previous literature performing photoactivation and/or photobleaching with LLSM noted above.

Response: We have added the related references as ref 18,19, and 20.

2. For the directed migration experiment, what are the associated imaging parameters (exposure time, number of steps, time between volumes). It is unclear if this is the same as previous experiments or not.

Response: We have added the associated imaging parameters in the material and methods part in line 414, "The camera settings for the directed cell migration experiment are the same (10 ms exposure time for 200 x 1024 pixels with 161 planes) for both colors 488 nm (activation) and 560 nm (imaging), where it takes ~ 3 s to finish 3D optogenetic imaging. Initially, the time interval between each volumetric image was set to 6 s until we observed cellular response or membrane ruffling (in this case, 60-time points). We then changed the time interval to 1 min to observe directed cell migration for 343-time points. Note that for cell manipulation, such as induced membrane ruffling and directed cell migration experiments, the subcellular responses of cells cannot be predicted. Therefore, we cannot program the SLM pattern time profile in advance. To demonstrate the capability of local activation, the photoactivation was conducted manually. After recording the first cellular response, the program was stopped, and the activation beam was moved to the desired area either by changing the SLM pattern or the x-galvo position. The program was resumed to record the cellular response. These processes were repeated until the end of the experiment."

3. In the final paragraph of the discussion, the authors reference a challenge in implementing 3D activation is the axial resolution. I am not sure what the authors are referring to at this point, as the axial resolution is the minimum distance at which two objects can be distinguish in the axial direction. I'm not sure how this connects to the 20 μ m referenced by the authors or is a limitation of the system. I intend for the above comments to be purely constructive to help improve the overall quality of the manuscript. I hope the authors and their families stay healthy during this time.

Response:We thank the reviewer pointed this out, and the reviewer is correct. We are sorry for the confusion. We rewrote this part as the following in line 321 "Another challenge in implementing 3D

activation in LLSM was the line confinement activation instead of a point in the propagation direction, which was about 20 μm at the present configuration in order to cover the whole cell imaging. Such confinement allows us to conduct local activation of molecules located on the apical or basal membrane by SLM manipulation but not enough to activate molecules within a specific organelle in the cells with the fixed annual mask. An aperture-free technique for the generation of activation beam and lattice beam at different beam lengths is required in the future. In summary, the 3D activation capability of our approach opens the opportunity for optogenetic activation of single cells in small animal models."

And many thanks for the reviewer's concern during this difficult time. We are really happy that we are still able to revise the manuscript during the COVID-19 pandemic.

Reviewers' comments:

Reviewer #1 (Remarks to the Author):

The authors have largely addressed my prior comments. The manuscript effectively demonstrates that a Bessel beam can be integrated into a lattice light sheet microscope for optogenetic activation. The authors demonstrate that changing the phase profile on the SLM can shift the Bessel beam relative to the lattice pattern used for imaging. Although the "3D" photoactivation is not very precise (poor photoactivatable confinement along the beam propagation axis), I think the revised title and claims are enough to address this from the prior submission. The authors then characterize their tool with optogenetic probes that induce fluorophore clustering and membrane ruffling in cells. Although the biological applications have not revealed any new or novel findings, I do believe the technological contribution has been adequately described and characterized. I have only a few remaining comments on the revised manuscript:

1) Line 25: "better spatiotemporal control of photoactivation". I'd suggest removing "better". It's not clear what "better" is being compared to (better than what?) and such a claim would require quantitative comparisons which were not performed in this manuscript.

2) Lines 68-69: "In addition, the 3D nature of LLSM may allow local activation of biological processes in a specific subcellular compartment or organelle". But in the discussion, the authors state (Lines 325-326): "SLM manipulation, but not enough to activate molecules within a specific organelle in the cells". It seems the discussion contradicts the claims in the introduction. I'd suggest removing above-referenced sentence in the introduction.

Reviewer #2 (Remarks to the Author):

This is the second set of review.

From previous version, positioning of this project has been well clarified (according to the existing scientific work).

While some sentences of the abstract, introduction and figure caption has been modified, the article would still benefit from another round of proofreading (shorter/clearer sentences, and typos for exemple).

Considering comments made previously, one main point remains unanswered (and underlined by reviewer 2 and 3):

this article lacks statistics on experiments made with ISH and EGFR domains. It is especially lacking A) one control: local experiment (activation on the border) for CRY2/CIBN and CRY2olig without ISH/EGFR to check in (the absence of) membrane activity; and B) a measure of how reproducible these experiments are i.e. to repeat the same type of experiment at least 5 times on different cells and mention how many time over these 5 experiments the authors observe the same phenotype (via quantitative measures).

This point seems major to me before publication.

Minor comments:

L21: "3D" optogenetic activation - optogenetic activation are always "3D", (except for TIRF), plus, from the results, you can not affirm you control resolution/activation along the Z' axis

L25: "better" than what?

L70: "a novel LLSM design" can be replaced by " the use of LLSM "

L87: as the word "3D" is not realistic, you can fuse 1) and 2) into "1) controllable subcellular optogenetic activation"

L180: "3D" can be suppress, as noone of the quantification deals with the Z' axis

L280: for the spreading of CRY2/CIBN optogenetic system, you can cite valon et al. biophysical

journal 2015

L294: you can mention the distance inbetween the 2 activations points at the origin of ruffling (Around 10um?)

L315: "in 3D" can be suppressed

L328: "3D" can be changed into subcellular

L333: Plasmid construction section: it is missing information of the origin of ISH and EGFR sequences, as well as whole plasmid map and final plasmid availability (on request? in addgene?)

L382-385: New sentence added is not clear

Figure 4 has to be highly modified. F4a axis XY' / XZ' are wrong! they are indeed XY / XZ ? according to the legend?

Figure S5/S6, activation regions have to be shown.

Figure S7 and corresponding movie: has the data looks very great (from the movie), it may require fluorescence and cell contour quantification

Figure S8 : figure caption is unclear and has to be completed.

Reviewer #3 (Remarks to the Author):

Summary:

The authors provide a revised manuscript with much additional data regarding the specific methods by which photomanipulation is performed and image quantification. Though a majority of my concerns regarding the manuscript were addressed, I still have a few reservations.

Major comments:

1. The authors note that an additional Arduino device was used to gate the excitation light when performing single Bessel beam photoactivation. This seems in conflict with the overall message that the original LLSM design was used for photomanipulation. I believe it should be made clear upfront that there are additional components necessary to perform the experiments described here. Additionally, a more detailed description of the Arduino-controlled shutter and the accompanying software should be provided to maintain reproducibility.

2. One major limitation appears to be that to change to photomanipulation location, the experiment must be paused, a new SLM pattern must be generated and uploaded, and then the experiment must be resumed. This places a large limitation on the speed of biological processes that can be studied and may further complicate monitoring timing of the experiment. This is an important point to discuss when describing limitation of this approach.

Reviewer #1 (Remarks to the Author):

Reviewer #1 (Remarks to the Author):

The authors have largely addressed my prior comments. The manuscript effectively demonstrates that a Bessel beam can be integrated into a lattice light sheet microscope for optogenetic activation. The authors demonstrate that changing the phase profile on the SLM can shift the Bessel beam relative to the lattice pattern used for imaging. Although the “3D” photoactivation is not very precise (poor photoactivatable confinement along the beam propagation axis), I think the revised title and claims are enough to address this from the prior submission. The authors then characterize their tool with optogenetic probes that induce fluorophore clustering and membrane ruffling in cells. Although the biological applications have not revealed any new or novel findings, I do believe the technological contribution has been adequately described and characterized. I have only a few remaining comments on the revised manuscript:

1) Line 25: “better spatiotemporal control of photoactivation”. I’d suggest removing “better”. It’s not clear what “better” is being compared to (better than what?) and such a claim would require quantitative comparisons which were not performed in this manuscript.

Response: We thank the reviewer’s suggestions. We have removed the word “better” in line 27 as “As a result, a Bessel beam as a stimulation source is integrated into the LLSM without changing the optical configuration, achieving spatiotemporal control of photoactivation.” in the revised manuscript.

2) Lines 68-69: “In addition, the 3D nature of LLSM may allow local activation of biological processes in a specific subcellular compartment or organelle”. But in the discussion, the authors state (Lines 325-326): “SLM manipulation, but not enough to activate molecules within a specific organelle in the cells”. It seems the discussion contradicts the claims in the introduction. I’d suggest removing above-referenced sentence in the introduction.

Response: We thank the reviewer’s suggestion, we have removed the sentence “In addition, the 3D nature of LLSM may allow local activation of biological processes in a specific subcellular compartment or organelle.” in the introduction.

Reviewer #2 (Remarks to the Author):

This is the second set of review.

From previous version, positioning of this project has been well clarified (according to the existing scientific work).

While some sentences of the abstract, introduction and figure caption has been modified, the article would still benefit from another round of proofreading (shorter/clearer sentences, and typos for exemple).

Considering comments made previously, one main point remains unanswered (and underlined by reviewer 2 and 3):

this article lacks statistics on experiments made with ISH and EGFR domains. It is especially lacking A) one control: local experiment (activation on the border) for CRY2/CIBN and CRY2olig without ISH/EGFR to check in (the absence of) membrane activity; and B) a measure of how reproducible these experiments are i.e. to repeat the same type of experiment at least 5 times on different cells and mention how many time over these 5 experiments the authors observe the same phenotype (via quantitative measures).

This point seems major to me before publication.

Response: We thank the reviewer's suggestion. In the revised manuscript, we have added more experimental results to demonstrate the reproducibility of our technique. As suggested by the reviewer, we conducted control experiments without ISH/EGFR. Five different cells were activated by 488 nm laser light at the borders of the cells. The time-lapse images of these five cells after the photostimulation are shown below as row (A) to row (E) where the activation areas are indicated by the yellow dotted lines. The time intervals between each image were of 15 or 20 minutes. In these control experiments, we observed that without ISH/EGFR, photostimulation only induced local membrane ruffling. No considerable cell migration was recorded during the entire imaging period, where time-lapse images were acquired every 10 s for 60 ~ 80 minutes.

We also conducted a control experiment for the optically induced clustering using CRY2olig without ISH/EGFR. The time-lapse images of photoactivation of a cell transfected with CRY2olig-mRuby3 without ISH/EGFR are shown below. It can be seen that the photoactivation induced oligomerization CRY2olig-mRuby3. The number of clusters increased over time as we continuously activated the cell. At the end of the experiment, most of the CRY2olig-mRuby3 molecules formed clusters making it difficult to visualize the boundary of the cell. During the experiment, no cell migration was observed

To demonstrate the reproducibility of the photostimulation on the membrane activity, we conducted six photostimulation experiments on six different cells using CRY2-CIBN with different cell morphology and expression level. The results of these experiments are shown below. For cell1 - cell3 (with typical morphologies and expression levels), the images of the cells before and after activation are shown in (A) and (B) respectively, where the 488 nm activation beam is indicated by the cyan line. To observe the membrane activity, the enlarged time-lapse images of the yellow dotted box in (A) and (B) are shown in (C) and (D), respectively. Before activation, there are weak local membrane activities. After Bessel beam activation, T=0 in (D), the rapid membrane rufflings resulted in strong fluorescence intensity variations as indicated by the yellow arrows. The morphology of cell4 is very different from cell1-cell3, where the cell adhesion to the glass surface resulted in fiber-like structures. Before the activation, there were some membrane activities on the top area of images (C). However, after activation at T=0 shown in (D), the membrane ruffling slowly changed the morphology of the cell. The sharp edge smoothed out within 1200 s due to vigorous membrane activities. The cell took some time to reshape the membrane due to the photoactivation and followed by the rapid membrane activities. To see how the photoactivation works for cells with a low expression level of CRY2-CIBN, we imaged two cells (cell 5 and cell6) with weak fluorescence intensities. As one can see with the same amount of energy for photoactivation, these two cells exhibited slower responses to the continuous activation than cell1-cell3 with response times up to several minutes. All the timestamps are in the unit of second.

From these experiments, we demonstrate that our technique is reproducible for photoactivation. However, it is pretty difficult to provide a quantitative analysis of the membrane activities or migration due to the transfection efficiency, the morphology of the cell shape, and how the cells attach to the coverslip. In the revised manuscript, we adapted reviewer 3's suggestion that the measurements and quantification of the membrane activities and clustering response are simply observations from individual examples instead of conclusions from repeated studies. In the revised manuscript on page 17, line 305. We have added, "Note that due to the individual cultured cells with different transfection efficiency, morphologies, and adherence to the coverslip, the quantified measurements in this study are based on the observations from individual examples."

Minor comments:

L21: "3D" optogenetic activation - optogenetic activation are always "3D", (except for TIRF), plus, from the results, you can not affirm you control resolution/activation along the Z' axis

Response: We thank the reviewer's suggestion. We have revised this sentence in Line 21 as "Lattice lightsheet microscopy (LLSM) featuring three-dimensional (3D) recording is modified to manipulate cellular behavior with subcellular resolution through optogenetic activation."

L25: "better" than what?

Response: As suggested by the reviewer, we have removed the word "better" in Line 25.

L70: "a novel LLSM design" can be replaced by " the use of LLSM "

Response: As suggested by the reviewer, we rewrote the sentence as, "In this study, we report the use of LLSM for optogenetic experiments."

L87: as the word "3D" is not realistic, you can fuse 1) and 2) into "1) controllable subcellular optogenetic activation"

Response: As suggested by the reviewer, we rewrote the sentence in line 88 as, "To achieve 3D cellular manipulation in LLSM, we demonstrated that our approach offers the advantages over original LLSM including controllable subcellular optogenetic activation and long-term cell manipulation."

L180: "3D" can be suppress, as noone of the quantification deals with the Z' axis

Response: As suggested by the reviewer, we removed the word "3D" in Line 184

L280: for the spreading of CRY2/CIBN optogenetic system, you can cite valon et al. biophysical journal 2015

Response: We thank the reviewer's suggestion, we have added the reference as reference 31 in Line 193 page 11.

L294: you can mention the distance inbetween the 2 activations points at the origin of ruffling (Around 10um?)

Response: We thank the reviewer's suggestion, we rewrote the sentence as followings in the revised manuscript in Line 295, "two independent membrane ruffling responses with a separation distance of 15 um could be triggered in the same cell with 1 nW Bessel fan activation at 488 nm (Figure S7)"

L315: "in 3D" can be suppressed

Response: We thank the reviewer's suggestion, we have removed " in 3D" in line 313. : As suggested by the reviewer, we have changed "3D" to "subcellular". In line 317, we have revised the sentence as

“ Therefore, the optogenetic molecules located on the membranes could be activated locally at the apical or basal membranes to achieve a controllable optogenetic system in subcellular fractions.”

L328: "3D" can be changed into subcellular

Response: As suggested by the reviewer, we have changed “3D” to “subcellular” in line 333.

L333: Plasmid construction section: it is missing information of the origin of ISH and EGFR sequences, as well as whole plasmid map and final plasmid availability (on request? in addgene?)

Response: The plasmids shown in the manuscript were constructed in our lab. The followings are the whole plasmid map and the plasmids are available upon request.

In the revised manuscript, we added the following sentence in line 369 “All aforementioned plasmids are available upon request.”

L382-385: New sentence added is not clear

Response: The added sentence was included to explain how to measure the laser power beyond the detection limit of the power meter as requested by reviewer 3. In the revised manuscript we rewrote the sentence in line 389 as “ The detection limit of the laser power meter is 1 nW. To obtain laser power of 0.5 nW, the laser power was first set at 5 nW according to the power meter and then attenuated linearly by a combination of neutral density filter and acousto-optic tunable filter (AOTF).”

Figure 4 has to be highly modified. F4a axis XY' / XZ' are wrong! they are indeed XY / XZ ? according to the legend?

Response: We thank the reviewer’s suggestion, we have revised the Fig 4a axis to the correct labels in XY' view and XZ view.

Figure S5/S6, activation regions have to be shown.

Response: We thank the reviewer’s suggestion, we have marked the activation regions in the revised Figures S5 and S6.

Figure S7 and corresponding movie: has the data looks very great (from the movie), it may require fluorescence and cell contour quantification

Response: We thank the reviewer’s suggestion. Yes, indeed, we are trying to demonstrate the capability of our technique to optically trigger the membrane activity in these movies, which display the membrane responses activated by single beam activation at different planes and imaged by the lattice beam excitation. We have added the contour quantification and fluorescence changes for the following 5-time points (stage 1 to 5) as follows. To show the morphological change induced by the photostimulation, the cell widths, and the integrated intensity are calculated with respect to different stages specified in Figure S7(a). A region of interest with the dimension of $3 \times 50 \mu\text{m}$ overlapped with the activation area (indicated by the blue solid line in (a)) is used to calculate the cell width and the integrated intensity. A time delay of 60 seconds after the stimulation is used for a significant change of the aforementioned physical parameters to be measured. As shown in (b), the local width measured at positions 1 & 2 increases as activated by stimulations and recovers to a smaller value after resting. After a subsequent global stimulation, both interaction sites' width increases again. Similarly, the integrated intensity of the specified activation areas responds to the local stimulations shown in (c). The intensity does not continue to increase upon the global stimulation, suggesting an even distribution of CRY2 over the entire cell. In Figure S7(d), the cell contour length is plotted with respect to the time. The cell contour is segmented by hierarchical k-means clustering and the contour length is smoothed by the Savitzky-Golay filter (25 points). It is noted that the cell contour length responds to the stimulation with a time delay of about tens of seconds to one minute.

The figure captions were changed to “Figure S7. (a) Subcellular activation of the cell expressing CRY2mCherryiSH-p2a-CIBNcaax. The top and bottom images are the enlarged views of the top and the

bottom activation areas (white dashed box in the middle images). The stimulating beams are illustrated in cyan. The widths of the cell at the stimulated regions are plotted in **(b)** and the integrated intensity associated with the activation areas is plotted in **(c)** at different stages of the stimulation experiment as shown in **(a)**. In **(c)**, the cell contour length is plotted with the time. The timepoints that the cell is subjected to stimulation are indicated by the arrow. Scale bar 10 μm “ In the revised manuscript, in line 199, we added, “To demonstrate that membrane ruffling can be induced with subcellular resolution, two areas within a cell that were separated by 15 μm were illuminated sequentially by the activation beam (regions 1 and 2 in Figure S7), and membrane ruffling at each location could be seen through quantification by fluorescent signals after stimulation.” .

Figure S8 : figure caption is unclear and has to be completed.

Response: We thank the reviewer’s suggestion. We have added more descriptions in the figure caption of Figure S8, as “Figure S8. Quantification of the membrane protrusion and retraction during the guided cell migration. A polar coordinate centered at the geometric center is used to quantify the amount of protrusion and retraction of the cell before and after photoactivation. (a) After determining the geometric center of the cell at each time point, the cell is divided into 24 sections with a separation of 12.5 degrees. The protrusion and retraction vectors are the vectors pointed outward from the geometric center and the vectors pointed inward to the center respectively. The magnitude of the vectors is calculated as the difference between the cell area within the section to the previous time point. The magnitude is normalized from -150 to 150 (arbitrary unit), which is indicated in the radial axis of the radar plots. In (b), the cell morphology before the stimulation is quantified with the radar plot. As a comparison, figure (c) indicates a change of cell polarity with a highly directional distribution of the protrusion and retraction vectors after the stimulation.”

Reviewer #3 (Remarks to the Author):

Summary:

The authors provide a revised manuscript with much additional data regarding the specific methods by which photomanipulation is performed and image quantification. Though a majority of my concerns regarding the manuscript were addressed, I still have a few reservations.

Major comments:

1. The authors note that an additional Arduino device was used to gate the excitation light when performing single Bessel beam photoactivation. This seems in conflict with the overall message that the original LLSM design was used for photomanipulation. I believe it should be made clear upfront that there are additional components necessary to perform the experiments described here. Additionally, a more detailed description of the Arduino-controlled shutter and the accompanying software should be provided to maintain reproducibility.

Response: We thank the reviewer's suggestion. Indeed, to perform single Bessel beam photoactivation as shown in Figures 2(a) and 3, we need another clock to control the shutter in front of the excitation laser. In line 159 of the revised manuscript, we rewrote the sentence as "With the help of an additional clock device in LLSM, the single Bessel beam activation could be achieved by switching activation wavelength 488 nm at a given frame within a 3D stack scan (Figure 2a single Bessel Beam) where the timing diagram indicates the activation beam is switched on once at a specific z step during volumetric imaging.". In the method part line 403 of the revised manuscript, the information about Arduino device is added as "controlled by a homemade Arduino device (Arduino Uno) and Labview program to synchronize the image acquisition software, SPIM." A microcontroller board Arduino Uno (ver R3, <https://www.arduino.cc/>) was used as a counter that received and counted TTL signals from the FPGA board and generated a TTL pulse at a specified time. A Labview program was used to communicate with SPIM to achieve the selected single Bessel beam activation.

2. One major limitation appears to be that to change to photomanipulation location, the experiment must be paused, a new SLM pattern must be generated and uploaded, and then the experiment must be resumed. This places a large limitation on the speed of biological processes that can be studied and may further complicate monitoring timing of the experiment. This is an important point to discuss when describing limitation of this approach.

Response: We thank the reviewer's suggestion. We have added the following sentence in the discussion to explain the limitation of our technique in line 326, "However, the integration of these techniques into SLM based LLSM may be complicated, especially for altering the activation positions during the image acquisition with present acquisition software."

REVIEWERS' COMMENTS:

Reviewer #2 (Remarks to the Author):

This is the third set of reviews,

Major concerns:

As previously written : while some sentences of the abstract, introduction and figure caption has been modified, the article would still benefit a lot from another round of revision to increase its clarity, especially from a massive rewriting of the abstract+introduction.

Moreover, I am still not convince by the figures/quantifications made on the optogenetic experiments, which are lacking single clear messages. I agree that cell responses during optogenetic experiments can be highly diverse, however, I am convince that there are ways to extract valuable information, either by increasing the amount of data, or by doing simpler/better quantifications.

Minor concerns:

I thank the authors for the clarification made with the control conditions (of the optogenetic experiments).

I thank the author for modifying the previously observed-by-the reviewers typos. However, it remains a lot of them which require in deep proofreading.

Reviewer #2 (Remarks to the Author):

This is the third set of reviews,

Major concerns:

As previously written : while some sentences of the abstract, introduction and figure caption has been modified, the article would still benefit a lot from another round of revision to increase its clarity, especially from a massive rewriting of the abstract+introduction.

Response: We thank the reviewer's comments. In the revised manuscript, we rewrote the abstract to help readers better understand our technique and findings. To make our point clear, we called our technique as optoLLSM which highlights the strength of our technique for the subcellular optogenetic activation. In the revised manuscript, the abstract has been rewritten as followings,

"Lattice lightsheet microscopy (LLSM) featuring three-dimensional recording is improved to manipulate cellular behavior with subcellular resolution through optogenetic activation (optoLLSM). A position-controllable Bessel beam as a stimulation source is integrated into the LLSM to achieve spatiotemporal photoactivation by changing the spatial light modulator (SLM) patterns. Unlike the point-scanning in a confocal microscope, the lattice beams are capable of wide-field optical sectioning for optogenetic activation along the Bessel beam path. We show that the energy power required for optogenetic activations is lower than 1 nW (or 24 mWcm⁻²) for time-lapses of CRY2olig clustering proteins, and membrane ruffling can be induced at different locations within a cell with subcellular resolution through light-triggered recruitment of phosphoinositide 3-kinase. Moreover, with the epidermal growth factor receptor (EGFR) fused with CRY2olig, we are able to demonstrate guided cell migration using optogenetic stimulation for up to 6 h, where 463 imaging volumes are collected, without noticeable cellular damages."

To improve the clarity of our manuscript, we make following changes in the revised manuscript:

All the words about light sheet, or light-sheet → "lightsheet"

All the words about optically induced clustering for CRY2 proteins → "CRY2olig"

All the figures have brief titles.

In line 1, optogenic → "optogenetic"

In line 39, three-dimensional → "3D"

In line 70, "called optoLLSM" was added ; our system → "optoLLSM"

In line 71, In our system, the LLSM beams is used to monitor cellular behavior, → "For optoLLSM, the lattice beams is used to monitor cellular behavior."

In line 75, our system → "optoLLSM"

In line 76, our technique → "optoLLSM"

In line 77, we conducted several cell-migration-related optogenetic experiments in our system, “in our system” was removed,

In line 86, “optoLLSM” was added

In line 92, To conduct an optogenetic experiment in LLSM , our system consists of a large parallel array of coherently interfering Bessel beams.... → “The optoLLSM consists of a large parallel array of coherently interfering Bessel beams....”

In line 135, the lattice beam could be shifted in the propagation direction → “the lattice beam could be repositioned in the propagation direction”

In line 156, To further demonstrate the capability of our system to conduct 3D activation → “To further demonstrate the capability of optoLLSM to conduct 3D activation”

In line 182, These findings clearly demonstrate that LLSM could be easily modified to conduct an optogenetic experiment → “These findings clearly demonstrate that optoLLSM could conduct an optogenetic experiment”

In line 186, the sentence “The next challenge of using LLSM for optogenetic application was to induce optogenetic responses that had subcellular resolution” was deleted

In line 227, Long-term manipulation of cell migration on lattice lightsheet microscope → “Long-term manipulation of cell migration by optoLLSM”

In line 242, LLSM → “optoLLSM”

In 248, LLSM → “optoLLSM”

In line 253, the combination of LLSM and Bessel fan → “optoLLSM with Bessel fan”

In line 259, In optogenetic LLSM experiments → “In optoLLSM experiments”

In line 335, the subcellular activation capability of our approach → “the subcellular activation capability of optoLLSM”

Moreover, I am still not convince by the figures/quantifications made on the optogenetic experiments, which are lacking single clear messages. I agree that cell responses during optogenetic experiments can be highly diverse, however, I am convince that there are ways to extract valuable information, either by increasing the amount of data, or by doing simpler/better quantifications.

Response: We thank the reviewer’s comments for pointing out the importance of the amount of optogenetic data and image quantifications. We agreed that a good analysis would be the key to extract useful information from the three-dimensional LLSM images. Also, a considerable amount of data would be helpful to reduce the variation between experiments and provide a factual data basis for statistics.

The experiments conducted in this study heavily relied on manual control of the microscope hardware and image analysis. Certainly, we would like to construct a turn-key system for LLSM and make it available for the biological community. Although the commercial version of LLSM is on the market from Zeiss company, there are still rooms needed to be improved for the different applications such as optogenetic application as shown in the manuscript. We believe that our invention for the light-trigger experiments

will benefit the LLSM users in the community, and probably for those who use the home-built optical microscope systems in the labs. The technique could evolve to fit the need for other biological applications. It may take some time and more validations.

In addition to the technical improvement of optoLLSM, there are several useful messages from our manuscript for the research community in biology. Firstly, in order not to get the cross-activation by the imaging excitation channel, we constructed a bright fluorescence probe fused with CRY2 system to minimize the energy dose required for 3D imaging acquisition, resulting in no activation from imaging wavelength. Furthermore, without cross-activation, we have chances to guide the moving cell for a long period of time (~ 6hrs with second resolution for 3D recording). Otherwise, the photobleaching and potential cross-activation will occur during the imaging acquisition.

In order to tackle the through-put problem for the described tool, we are making better temperature control system and CO₂ environment for living cell experiment. From biological sides, we have constructed the stable cell lines for equal expression level. In the near future, we could scale up the data number and speed up the imaging analysis and we hope that the reviewer could understand our current limitation.

Minor concerns:

I thank the authors for the clarification made with the control conditions (of the optogenetic experiments).

Response: We thank the reviewer's suggestions and the chance for us to improve our data presentation.

I thank the author for modifying the previously observed-by-the reviewers typos. However, it remains a lot of them which require in deep proofreading.

Response: We thank the reviewer for pointing out the typos. We have revised our manuscript accordingly, as shown in the aforementioned reply. We also carefully went through the manuscript to correct all typos.